# The Effect of Carbohydrate Intake on Strength and Resistance Training Performance: A Systematic Review

**DOI:** 10.3390/nu14040856

**Published:** 2022-02-18

**Authors:** Menno Henselmans, Thomas Bjørnsen, Richie Hedderman, Fredrik Tonstad Vårvik

**Affiliations:** 1The International Scientific Research Foundation for Fitness and Nutrition, David Blesstraat 28HS, 1073 LC Amsterdam, The Netherlands; hurricanefitnessireland@gmail.com (R.H.); ftvaarvik@gmail.com (F.T.V.); 2Department of Sport Science and Physical Education, Faculty of Health and Sport Sciences, University of Agder, 4630 Kristiansand, Norway; thomas.bjornsen@uia.no

**Keywords:** resistance exercise, carbohydrate intake, muscle strength, performance

## Abstract

High carbohydrate intakes are commonly recommended for athletes of various sports, including strength trainees, to optimize performance. However, the effect of carbohydrate intake on strength training performance has not been systematically analyzed. A systematic literature search was conducted for trials that manipulated carbohydrate intake, including supplements, and measured strength, resistance training or power either acutely or after a diet and strength training program. Studies were categorized as either (1) acute supplementation, (2) exercise-induced glycogen depletion with subsequent carbohydrate manipulation, (3) short-term (2–7 days) carbohydrate manipulation or (4) changes in performance after longer-term diet manipulation and strength training. Forty-nine studies were included: 19 acute, six glycogen depletion, seven short-term and 17 long-term studies. Participants were strength trainees or athletes (39 studies), recreationally active (six studies) or untrained (four studies). Acutely, higher carbohydrate intake did not improve performance in 13 studies and enhanced performance in six studies, primarily in those with fasted control groups and workouts with over 10 sets per muscle group. One study found that a carbohydrate meal improved performance compared to water but not in comparison to a sensory-matched placebo breakfast. There was no evidence of a dose-response effect. After glycogen depletion, carbohydrate supplementation improved performance in three studies compared to placebo, in particular during bi-daily workouts, but not in research with isocaloric controls. None of the seven short-term studies found beneficial effects of carbohydrate manipulation. Longer-term changes in performance were not influenced by carbohydrate intake in 15 studies; one study favored the higher- and one the lower-carbohydrate condition. Carbohydrate intake per se is unlikely to strength training performance in a fed state in workouts consisting of up to 10 sets per muscle group. Performance during higher volumes may benefit from carbohydrates, but more studies with isocaloric control groups, sensory-matched placebos and locally measured glycogen depletion are needed.

## 1. Introduction

Dietary carbohydrates can enhance performance in endurance sports, as they are the preferred muscular energy substrate at moderate to high intensities [1]. There is less research about carbohydrate requirements for strength training, such as Olympic weightlifting, powerlifting and bodybuilding. Resistance training is metabolically distinct from endurance training and leads to different training stimulus and adaptive responses, so it may have different carbohydrate requirements [2].

Carbohydrates can be stored as glycogen in the liver (approximately 80–120 g) and muscles (approximately 350–700 g) [3]. Muscle contractions during both low- and high-load resistance training rely primarily on the anaerobic glycolysis pathway for energy, as there is insufficient oxygen to rely purely on the aerobic system and fatty acids to provide energy sufficiently rapidly [4,5,6]. Hence, glycogen depletion could limit performance. Glycogen is localized in three main subcellular compartments within the muscle cell; under the sarcolemma, intermyofibrillar between the myofibrils and intramyofibrillar within the myofibrils [7]. Glycogen depletion can occur locally in these subcellular glycogen departments after resistance training, even if whole-muscle glycogen levels are only partially depleted [8]. Excessive glycogen depletion can contribute to muscle fatigue by lowering ATP synthesis [7,9], and possibly also by lowering muscle excitation and impairing calcium release from the sarcoplasmic reticulum [10,11]. Endurance exercise in a significantly glycogen-depleted state can also increase protein oxidation and reduce muscle protein synthesis [12,13]; however, low pre-exercise glycogen availability has not been found to significantly affect anabolic signaling or muscle protein synthesis after strength training [12,13]. While low glycogen availability per se may not be detrimental for muscle anabolism, it can impair strength performance and training volume [14,15]. In addition, strength-trained individuals can achieve higher work outputs during exercise and have a greater capacity for glycogen storage compared to untrained individuals [16,17]. Thus, trained individuals may require higher carbohydrate intakes to optimize performance, although training status does not seem to influence the relative level of glycogen depletion after a given resistance training workout [18].

Previous reviews have recommended carbohydrate intakes of 8–10 g per kilogram of bodyweight per day (g/kg/day) during ‘heavy anaerobic exercise’ [16]. Others recommend 4–7 g/kg/day for strength athletes to optimize strength performance and hypertrophy [19,20]. These recommendations are not far from the common 6–12 g/kg/day recommendation for endurance athletes [1]. The average daily intake of carbohydrates in bodybuilders has been reported to range from 2.8 to 7.5 g/kg/day, compared to 4.2–8 g/kg/day in strength-athletes [20,21]. However, none of these recommendations or practices stem from a systematic literature review, only narrative reviews. Thus, this systematic review examines whether carbohydrate intake influences acute and longer-term strength training performance.

## 2. Methods

The present systematic review followed Preferred Reporting Items for Systematic Reviews and Meta-Analyses (PRISMA) guidelines [22]. We did not pre-register the present review, because the protocol did not fulfill the requirements for preregistration at Prospero, which state that they do not accept reviews assessing sports performance as an outcome. However, in retrospect, there are other options that we have used [23].

### 2.1. Search Strategy

A literature search was conducted in EBSCOhost within the MEDLINE and SPORTDiscus databases, in addition to the SciELO database. Search terms included a combination of Medical Subject Headings (MeSH terms) and free-test words consisting of the following keywords:

“(MH “Carbohydrates”) OR (“glycogen depletion” OR “high carbohydrate” OR “low carbohydrate” OR keto* OR (maltodextrin N2 (supplement* OR intake) OR (glucose N2 (ingestion OR intake OR supplement) OR (carbohydrate* N6 (intake* OR supplement* OR manipulat* OR consumption OR ingestion OR feeding OR restricti* OR diet OR drink OR breakfast)) AND MH “Resistance Training” OR MH “Weight Lift*” OR (isokinetic OR “strength training” OR “resistance training” OR “resistance exercise” OR powerlift* OR weightlift* OR “power lift” OR CrossFit) AND (MH “Muscle Strength”) OR (strength OR 1 RM OR performance OR failure OR power OR “total work” OR torque OR force OR volume)”. The search strategy for each database can be found in the Appendix A, gray literature (master theses, PhD dissertations and conference abstracts) were searched for with the same keywords in Google Scholar. The last literature search included publications up until the 1 January 2022 and was completed by RH and FTV.

### 2.2. Inclusion Criteria

Online published trials were included if they compared conditions with different carbohydrate intakes, including supplements, and measured dynamic resistance training performance as an outcome. Studies were categorized as either (1) acute carbohydrate manipulation (up to 24 h) or supplementation prior to strength tests, (2) exercise-induced glycogen depletion and carbohydrate manipulation prior to strength tests, (3) short-term carbohydrate manipulation of at least a day and up to a week prior to strength tests or (4) long-term changes in strength performance after more than a week of carbohydrate manipulation and strength training. Strength performance was measured in the form of maximal strength (1 repetition maximum (1 RM), isokinetic work or peak or average torque), repetitions to failure (within a single set, number of sets to predetermined repetition failure or total repetition volume when training to failure) or power (average or peak). Isometric strength measurements were only included if participants also performed dynamic strength measurements to ensure the assessed outcomes were practically relevant to strength trainees. In addition, sprinting, agility, jumping, short-distance running or Wingate performance were included as secondary outcomes if at least one of the primary outcomes were measured. Participants had to be healthy (i.e., free of chronic diseases) and below 60 years of age. Studies with concurrent training were included only if the endurance training was performed in a separate session. Papers in all languages were eligible. Congress abstracts were eligible for inclusion but presented in their own sections, not among the main findings. Letters were not included.

### 2.3. Study Selection and Data Extraction

Title and abstracts were screened by HR and FTV, followed by review of the full texts. Any disagreement between authors were discussed with all authors until a consensus was reached. Each of the included studies’ citations and related review articles were also screened for additional articles fulfilling the inclusion criteria. Data from each study were extracted to a spreadsheet, including (a) citation, (b) study design, (c) participant characteristics and sample size, (d) experimental details (including fed or fasted state and carbohydrate intake in acute and glycogen depletion studies and daily macronutrient intake in short- and long-term studies) and (e) results.

### 2.4. Quality Assessment

Study quality was assessed using the validated Tool for the Assessment of Study Quality and Reporting in Exercise (TESTEX) scale [24]. TESTEX is a 15-point assessment scale, consisting of 5 points for study quality and 10 for study reporting. Higher scores reflect better study quality and reporting. However, points 6C (exercise attendance), 7 (intention to treat analysis), 10 (activity monitoring in control groups) and 11 (progressive program) were excluded for the acute-, glycogen depletion- and short-term study categories, as these items did not apply in an acute context where participants do not follow a long-term diet and training intervention. Additionally, point 10 (activity of the control group) was excluded for the assessment of the long-term studies, as we only included studies with the same exercise intervention. For point 7 (intention to treat analysis), a point was given if there were no dropouts and therefore no need for an intention-to-treat analysis. For point 11 (consistent training intensity), a point was given if the participants were athletes or strength trainees following their regular program. Thus, the maximum scores for the acute and longer-term studies were 11 and 14, respectively. Furthermore, point 6C (exercise attendance) was not considered applicable for the long-term studies of shorter duration in which participants were not prescribed any training in between strength tests, so it was marked as “n/a”. The assessment was performed independently by FTV and TB. If any points were unclear, they were discussed until an agreement was reached. Based on the sum of the scores, studies were classified as having either excellent quality (acute studies: 10–11 points, long-term studies: 12–14 points), good quality (acute: 8–9, long-term: 10–11), fair quality (acute: 5–7, long-term: 7–9) or low quality (acute: <5, long-term: <7) [25].

## 3. Results

The literature search yielded a total of 504 papers after duplicate removal; 447 papers were excluded based on their title and abstract. After examining the 57 remaining full texts, 40 papers from the main search were included as well as four additional papers from reference lists, four theses/dissertations and one from authors’ previous knowledge on the topic (Figure 1), amounting to 19 acute, five exercise-induced glycogen depletion, seven short-term and 17 long-term studies. Additionally, four published abstracts were included (three acute and one short-term). Study quality in the acute, glycogen depleted and short-term studies was rated good (8 ± 1 points); longer-term study quality was rated fair (9 ± 1 points). See summary Table 1 and Table 2 and individual study results in Table A1 and Table A2 (Appendix B).

### 3.1. The Effect of Acute Carbohydrate Manipulation on Strength Training Performance

Nineteen publications met the inclusion criteria [26,27,28,29,30,31,32,33,34,35,36,37,38,39,40,41,42,43,44], summarized in Table 3 and Figure 2. Sixteen studies were crossover trials with an average of 11 ± 4 participants (median: 9) [26,27,28,29,30,31,34,35,36,37,38,39,41,42,43,44]; three studies were randomized controlled trials (RCTs) with an average of 11 ± 5 participants (median: 9) per group [32,33,40]. Participants were young, with an average age of 23 ± 2 years. Training status of the participants was categorized as untrained [32], physically active [42], CrossFit trained [38,44], recreationally trained [27,40,43] or strength-trained [26,28,29,30,31,33,34,35,36,37,39,41]. All participants were men except for 6 of the 17 participants in Fairchild et al. [27] and all 13 in Raposo [37].

The participants were in fasted or unspecified conditions in 13 studies before carbohydrate manipulation and the performance tests [27,29,30,31,32,33,36,37,38,39,41,42,44]; six studies were performed with both groups in a fed state (2.5–5 h) before performance was measured [26,28,34,35,40,43]. Only one of the studies was isocaloric [36]. Workout types included traditional strength training [26,29,30,31,32,33,34,35,37,39,40,41,42,43], CrossFit [38,44], circuit training [36], and isokinetic exercise [27,28]. Fifteen studies measured performance as repetitions to failure, sets to predetermined repetition failure, total training volume and/or total work (kJ) during traditional strength exercises [26,29,30,31,32,33,34,35,37,38,39,40,41,42,43] or CrossFit [36,44], of which one also included peak power [30] and two studies measured isokinetic performance (average and peak power, isokinetic peak or average torque, total and average work across sets) [27,28]. Four studies also measured secondary outcomes, such as agility and sprint time [29,36], jump distance and throwing performance [29] and work output measured as caloric expenditure during maximal effort rowing [38].

#### 3.1.1. Main Findings

In total, 11 of the 19 acute studies found no significant effect of carbohydrate intake on strength training performance [26,31,32,33,34,35,37,38,41,43,44]. Of the eight studies with a significant between-group effect, six favored the higher-carbohydrate condition: five studies reported more repetitions to failure/training volume [29,30,39,40,42] (but not peak power [30]) and one study reported greater isokinetic performance [28]. In these studies, the higher-carbohydrate conditions also had a higher energy intake, because the control conditions were either continuing their overnight fast [39], received a non-caloric placebo after an overnight fast [30] or in an unspecified state [42], received 5.5 g amino acids as placebo [29] or received a non-caloric placebo 3–5 h after a meal [28,40]. None of the isocaloric comparisons found the higher carbohydrate condition had greater performance than the lower carbohydrate condition [36,40,44].

Positive effects of carbohydrate intake were more prevalent when compared to fasts of four or more hours, but the effect of fasting duration was not clear. In the six studies in a relatively fed state with no more than five hours of fasting [26,28,34,35,40,43], three studies found no significant effects of carbohydrate intake after 2–4 h fasts [34,35,43], one found a non-significant trend for benefits in some of the outcome measurements after a four hour fast [26] and two studies found a positive effect of supplementing carbohydrates after a 3- or 5-h fast in some but not all of the tested outcomes [28,40]. In the 11 studies comparing carbohydrate intake to an overnight fasted state [27,29,30,31,32,33,37,38,39,41,44], one found a benefit of carbohydrates for both measured outcomes [39], two found a benefit for some but not all of the performance measurements [29,30], seven found no significant effects of carbohydrate intake [31,32,33,37,38,41,44] and one found a detrimental effect [27]. Two studies did not specify whether the subjects were fed or fasted [36,42].

Positive findings of carbohydrate intake compared to fasting are not necessarily indicative of a metabolic advantage of carbohydrate consumption. One study in resistance-trained men [41] found that the ergogenic effect of the higher carbohydrate condition was a placebo or at least non-metabolic effect: a carbohydrate-breakfast resulted in significantly more squat (but not bench press) repetitions to failure compared to a water-only control group but not compared to a flavor- and texture-matched placebo breakfast with only 29 kcal.

Two of the acute studies reported greater performance in the lower-carbohydrate conditions: one favored a non-caloric placebo over carbohydrate intake for isokinetic performance (when adjusting for baseline values) [27] and an isocaloric study favored a high-protein beverage to a high-carbohydrate beverage for aggregate performance on a battery of agility, push-ups and sprint tests, though not on any individually analyzed test [36].

#### 3.1.2. Carbohydrate Dosage and Training Volume

There was no dose-response effect of carbohydrate intake on performance. Significant effects were observed with dosages as low as 0.27 g/kg [42], 0.81 [40] and 1.5 g/kg [28,39], yet not found in five studies with 0.2–0.59 g/kg [31,32,35,38,44] or five studies at 0.89–1.5 g/kg [33,34,37,41,44] or 2 g/kg [43]. Krings et al. [29] studied the effect of 15 g/h, 30 g/h and 60 g/h of carbohydrates vs. placebo on strength training, running and jumping performance. Supplementing with carbohydrates significantly improved performance compared to placebo only in the bench press at all doses and for the 27-m sprint only at the 60 g/h dosage, without any dose-response effect. In fact, the 15 g/h group significantly outperformed the 60 g/h group for the bench press and total repetitions across all resistance training exercises. In absolute terms, the highest number of repetitions were achieved for the bench press, bent-over row, and triceps extension in the 15 g/h carbohydrate group and for the biceps curl in the 30 g/h carbohydrate group, none for the 60 g/h carbohydrate group. The two other studies comparing multiple doses of carbohydrates also found no effects of carbohydrate dosage on performance. Welikonich [40] had participants consume a pre- and intra-workout drink with either 60 g carbohydrate or 50 g carbohydrate and 14 g protein or placebo (21 kcal). The lower and higher carbohydrate drinks both improved leg press repetition performance identically (135 total repetitions) compared to the placebo drink. Maroufi et al. [44] found no significant difference in the total number of repetitions that could be completed during two CrossFit workouts by CrossFit athletes after consuming a non-caloric placebo or after consuming 45 g carbohydrates and protein or 67.5 g carbohydrates and 22.5 g protein.

Positive effects of higher carbohydrate intakes were more consistent in higher training volume workouts. In studies with performance tests consisting of more than 10 sets per muscle group (11–17 sets), significant positive effects of higher carbohydrate intake [28,40] or a trend thereof [26] were observed in three studies, whereas one study found no significant effects [33]. Out of 14 studies [27,29,31,32,34,35,36,37,38,39,41,42,43] with lower-volume performance tests (≤7 sets per muscle group), three studies [29,39,42] significantly favored the carbohydrate conditions, and two favored the lower-carbohydrate conditions [27,36].

#### 3.1.3. Results from Published Abstracts

Three published abstracts involving acute carbohydrate manipulation have been published. Two of those observed no effect of carbohydrate intake on multiple sets of squats to failure [45] or isokinetic work, power, fatigue and peak torque [46]. One study found a carbohydrate supplement consumed in a semi-fasted state resulted in more repetitions during the last set of leg presses, but there was no effect for total repetitions of either the bench press or leg press [47].

### 3.2. The Effect of Exercise-Induced Glycogen Depletion and Carbohydrate Manipulation on Acute Strength Training Performance

Six studies were included that measured strength training performance after exercise-induced glycogen depletion [15,48,49,50,51,52], summarized in Table 4 and Figure 3. All were crossover trials with an average sample size of 9 ± 4, consisting of recreationally active [15], high-intensity trained [51] or strength-trained individuals [48,49,50,52] with an average age of 24 ± 2 years. Five studies contained only men [48,49,50,51,52]; one study consisted of five men and one woman [15]. One study by Haff et al. measured glycogen depletion after nine sets of squats and three sets of isokinetic leg extensions and again after three more sets of maximal isokinetic leg extensions/flexions [52]. When training fasted, glycogen concentrations decreased by 19.2% after the squats, rising to 40.7% after the isokinetic exercise. When training after consuming 1 g/kg carbohydrate pre-workout and every 10 min intra-workout, the depletion levels were reduced to 15.2% and 26.5%, respectively. The five other studies did not measure glycogen levels but instead designed their exercise-induced glycogen depletion sessions to deplete glycogen stores by ~80% via bicycling based on previous work [15,48,50,51] or high-volume (15 sets) strength training similar to the aforementioned study by Haff et al. [49]. The low-carbohydrate conditions consisted of either continuing their overnight fast [50], ~1.2 g/kg carbohydrates in the hours between their depletion session and strength tests [49], normal/high carbohydrate intakes up until a three-hour fast [52] or a daily carbohydrate intake of ~0.4–1.8 g/kg [15,48,51]. Carbohydrate intakes in the high-carbohydrate conditions consisted of either 1.2 g/kg as a single [50] or repeated dosage totaling ~2 g/kg [49,52], 7.7 g/kg/day [48], or they were not reported [15,51]. The strength workouts included traditional strength training [48,49,50] and isokinetic exercise [15,51,52]. Performance measures consisted of total training volume [15,48,49] or total work (J) [51,52] in addition to power, force and velocity across sets [50] and peak and average torque [15,51,52].

#### 3.2.1. Main Findings

Three of the six studies favored the higher carbohydrate intake over the non-caloric conditions [15,49,50]. Compared to a non-caloric placebo or unspecified condition, two studies observed significantly more repetitions to failure in the high-carbohydrate conditions [15,49] and one study observed higher average power outputs [50]. The only calorie-matched experiment by Mitchell et al. [48] found no significant between-group differences in total training volume during 15 sets of quadriceps strength training (five sets each of squats, knee extensions and leg presses) at 15 RM repetitions failure. The glycogen depletion workout consisted of bicycling followed by 48-h carbohydrate intakes of 0.38 g/kg/day (32 g/day) vs. 7.65 g/kg/day (643 g/day). A similar trial by Symons and Jacobs [51] had trained men perform a glycogen depletion workout followed by 48 h of a lower-carbohydrate diet (1.8 g/kg/day, 140 g) compared to an unspecified higher-carbohydrate diet. Afterwards, they performed 50 maximal isokinetic leg extensions and isometric strength tests. There were no significant between-group differences in isometric strength, total work (J), neuromuscular fatigue or peak or average torque. In addition, no group-differences were observed in total work (J), peak or average torque compared to the non-caloric placebo condition in Haff et al. [52].

#### 3.2.2. Carbohydrate Dosage and Training Volume

Training volume did not clearly mediate the effect of carbohydrate intake on strength training performance. Positive effects were observed in workouts with 5–19 sets [15,49,50] yet not in a trial with 15 sets [48] or during three sets of 10 repetitions [52] or during 50 maximal isokinetic knee extensions [51]. Similarly, no dose-response of carbohydrate intake was evident, with a lack of an effect at 1.9 g/kg [52] and 7.7 g/kg/day [48] yet significant effects at intakes of 1.2–2.0 g/kg [49,50].

### 3.3. The Effect of Short-Term Carbohydrate Manipulation on Acute Strength Training Performance

Seven studies that investigated the short-term effect of a higher-carbohydrate vs. a lower-carbohydrate diet were included [53,54,55,56,57,58,59], summarized in Table 5 and Figure 4. These studies lasted for a duration of 48 h to 1 week (average 5 days). Four studies were crossover trials with an average of 15 ± 11 participants (median: 11) [54,55,58,59] and three studies were RCTs with an average of 19 ± 5 participants (median: 18) in each study [53,56,57]. Participants were young, with an average age of 26 ± 3. Three studies included only men [55,56,59], two studies consisted of only women [54,57] and two studies contained both sexes [53,58]. Training status was categorized as sedentary [57], recreationally active [54], strength-trained [55,58] bodybuilders [59], CrossFit athletes [53] or hockey athletes [56]. Daily carbohydrate intakes in the lower-carbohydrate conditions ranged from 31–346 g (average: 160 g) compared to 165–672 g (average: 390 g) in the higher-carbohydrate conditions. Three of the short-term studies did not prescribe resistance training within the study period [54,55,58]; in the other four studies, participants were instructed to continue their regular resistance training [59] or physical activity [57], or sport-specific training [56], or to complete a 1 week resistance training log and thus presumably continue their strength training routine [58]. Exercise test protocols included dynamic resistance exercises for the lower body [54,55,57,59] or the whole body [56,58] or CrossFit workouts [53]. Performance measurements included isokinetic lower-body strength tests [54,57], maximal strength tests (1 RM) [56,58], repetitions to failure, total training volume or work (J) [55,58,59], CrossFit performance [53], jump height [56,58], lower- or upper-body power [55,58] and a Wingate test [58].

#### 3.3.1. Main Findings

None of the five randomized studies found significant effects of carbohydrate intake on performance, including the only isocaloric study [53] and one of the two non-randomized crossover trials [59]. A crossover trial by Sawyer et al. [58] favored the lower-carbohydrate condition for some measures, but due to a lack of randomization or counterbalancing, it was confounded by a possible order/familiarity effect.

#### 3.3.2. Results from Published Abstracts

One published abstract involving short-term carbohydrate manipulation was included [60]. In this study, 7 days of a carbohydrate loading diet did not increase resistance training performance compared to a control condition.

### 3.4. The Effect of Longer-Term Carbohydrate Diets and Strength-Training on Changes in Strength Performance

Seventeen studies that examined the long-term effects of different carbohydrate intakes on resistance training performance met the inclusion criteria [61,62,63,64,65,66,67,68,69,70,71,72,73,74,75,76,77], summarized in Table 6 and Figure 5. Study durations ranged from three weeks to three months (average and median: 8 weeks). Four studies were crossover trials with an average of 23 ± 7 participants (median: 11) [61,64,70,75]; 10 studies were RCTs [62,65,66,67,68,71,73,74,76,77] and three controlled trials [63,69,72] with an average of 37 ± 56 participants (median: 21) in each study. Participants were young, with an average age of 29 ± 8, and the majority were men, but two studies consisted of only women [66,73] and six studies contained both sexes [61,62,63,69,72,75]. Training status was categorized as untrained [66], active [74], military trained [71,72], strength-trained [63,64,65,73,77], bodybuilders [76], powerlifters or weightlifters [61,68], CrossFit athletes [62,69,75] or other athletes [67,70].

Daily carbohydrate intakes ranged from 15 to 347 g (average: 100 g, median: 44 g) in the lower-carbohydrate groups, corresponding to 3–52% (average: 17%, median: 9%) of total caloric intake, compared to 82 to 758 g (average: 330 g, median: 275 g) in the higher carbohydrate groups, corresponding to 15–70 (average: 49%, median 47%) of total caloric intake.

To improve dietary adherence, participants were either instructed to follow prescribed diets [61,71,76], they were provided with a list of foods to eat [67] or menus [75], they had frequent meetings with a dietitian [63,66,72], they had to deliver food records [62,65,68,69,70,74], they had frequent coaching and were given meal plans [77] or pre-cooked meals [72], or they were supervised and provided with packed meals [64]. Six of the thirteen ketogenic diet studies (<100 g/day carbohydrate) also monitored and confirmed ketosis with measuring ketone levels [65,72,73,75,76,77].

Exercise protocols during the interventions included dynamic resistance training with full-body workouts [63,72] or body part split workouts [65,68,71,73,74,77], maintenance of non-specified/habitual resistance training [64], circuit training [66], CrossFit [62,69], powerlifting and weightlifting [61], athletic sport-specific exercises in addition to strength exercise [67] or high-level gymnastics training [70]. In the studies where participants did traditional resistance training, the training consisted of 1–25 repetitions for 1–5 sets with a moderate to high load (>50% 1 RM) 2–4 times per week [61,63,65,67,68,71,72,73,74,77]. Six studies did not prescribe a training protocol but instructed the participants to continue their habitual resistance training program [61,64,76], CrossFit routine [69,75] or gymnastic training schedule [70]. Performance measurements included isokinetic knee flexion and extension strength tests in two studies [64,71], 1 RM strength tests in twelve studies [61,64,65,66,68,69,71,72,73,74,76,77] 10 RM performance in one trial [63], repetitions to failure or total training volume in four studies [64,66,69,70], CrossFit performance in two studies [62,75] and grip strength and various predominantly anaerobic athletic tests in six studies [62,65,67,69,72,73].

#### Main Findings

In total, 15 out of 17 studies found no significant effects of carbohydrate intake on strength training performance or strength development, including the eight studies with isocaloric and isonitrogenous comparison groups [61,62,63,64,65,75,76,77]. The single study favoring the higher-carbohydrate condition was Vargas-Molina et al. [73], who found significantly greater 1 RM squat and bench press strength development but not countermovement jump height after an 8-week higher-carbohydrate diet (282 g) compared to a low-carbohydrate (39 g) ketogenic diet in strength-trained women. The high-carbohydrate diet group also consumed more total calories (1980 kcal vs. 1710 kcal), resulting in fat loss in the ketogenic group but not the higher-carbohydrate group. The other 12 ketogenic studies found no significant between-group performance differences; seven when groups were isocaloric [61,62,63,65,75,76,77], one when the ketogenic diet group was lower in calories [68], and in four studies that did not report if energy intake significantly differed between groups [67,69,70,72]. The single study favoring the low-carbohydrate ketogenic diet group was by Rhyu and Cho [67], who found a lower Wingate “fatigue index” compared to a non-ketogenic diet group; however, there were no significant between-group differences in Wingate, maximal strength, 100 m-sprint or broad jump performance.

## 4. Discussion

The majority of 39 out of 49 studies, including all 16 isocaloric comparisons, found no significant benefits of carbohydrate manipulation on strength training performance. Similarly, three of the four published abstracts found no significant effects of carbohydrate intake on strength training performance, whilst one found higher repetition performance in one set for one exercise in the higher carbohydrate group but not for total repetition volume for either measured exercise. Ten studies found that carbohydrate consumption might enhance strength training performance in specific contexts, notably for otherwise fasted training, workouts with volumes over 10 sets per muscle group and bi-daily workouts. Four studies favored the lower carbohydrate condition, but these benefits may have been attributable to study confounders, such as a higher protein intake, rather than carbohydrate restriction.

### 4.1. The Effect of Acute Carbohydrate Manipulation on Strength Training Performance

Out of the 19 included studies, 11 studies found no significant effects of carbohydrate intake on strength training performance [26,31,32,33,34,35,37,38,41,43,44], six studies found significantly greater performance in the higher-carbohydrate conditions [28,29,30,39,40,42] and two studies found significantly greater performance in the lower-carbohydrate conditions [27,36].

Since there is no established mechanism by which carbohydrates would acutely impair performance, the two studies finding negative effects of carbohydrates may be type I errors. Fairchild et al. [27] found greater average and peak torque during seven sets of 3 RM isokinetic leg extensions after a non-caloric placebo than after consuming 1.1 g/kg carbohydrate. Both were consumed after an overnight fast by strength-trained men and women. However, the authors interpreted their findings primarily as a null effect rather than an effect favoring lower-carbohydrate intakes. Lynch [36] compared a 0.81 g/kg high-carbohydrate beverage to an isocaloric high-protein beverage in strength-trained men during a double-blinded, randomized, controlled, crossover trial. The participants performed a 15–18-min high-intensity resistance training workout followed by the drinks and 2 h rest before a test workout of agility *T*-tests, push-ups to failure and 40-m sprinting. The high-protein drink resulted in greater performance on aggregate test performance, though not on any individually analyzed test. Since amino acids are theoretically unlikely to aid strength training performance via mechanisms not shared by carbohydrates (e.g., providing glucose via gluconeogenesis or insulin-mediated suppression of protein breakdown), protein’s positive effect may have resulted from greater muscle protein synthesis and subsequent recovery [12,13,78] in between the two workouts, rather than an acute ergogenic effect per se. Thus, Lynch’s [36] findings may be interpreted as a null effect of carbohydrate intake and a positive effect of protein intake, not a positive effect of carbohydrate restriction. In the two other isocaloric comparisons with protein intake, protein intake had similar effects on performance as carbohydrate [40,44].

The lack of acute effects of carbohydrate intake on acute strength training performance in most studies can be understood based on its partial level of whole-muscle glycogen depletion, particularly in the lower-volume studies. While high-intensity, anaerobic exercise may seem to rely greatly on carbohydrates, the total cumulative demand may not easily exceed bodily stores during resistance training. Muscle contractions during both low- and high-load resistance training rely primarily on the anaerobic glycolysis pathway for energy, as there is insufficient oxygen to rely purely on the aerobic system and fatty acids to provide energy sufficiently rapidly [4,5]. Glucose and glycogen are therefore primary energy substrates to fuel anaerobic exercise such as resistance training [79,80]. Lambert and Flynn [6] estimated the glycolytic system to provide 82% of the adenosine triphosphate (ATP) demand of a set of biceps curls at 80% of 1 RM to muscular failure. However, the total energy expenditure of strength training is generally lower than that of endurance-type activities [81]. One reason for the lower energy expenditure is the involvement of eccentric muscle contractions, which require relatively little energy expenditure compared to concentric muscle contractions, because they involve biomechanical rather than chemical cleaving of actin-myosin cross-bridges [82]. Second, strength training exercise is very intermittent with often 1–3 min of rest after each set of exercise [83]. These rest periods allow the aerobic system, which can be fueled by fatty acids rather than carbohydrates, to contribute a considerable portion of the workout’s total energy expenditure: estimates range substantially from 20 to 70% of resistance training energy expenditure [84,85]. Moreover, many sets are short enough that the creatine phosphate system, relying on creatine and stored ATP, can contribute approximately 16% of energy demands of a set of high-intensity resistance training, with 31% contributions being possible with shorter-duration anaerobic efforts, such as Wingate tests [6,86,87]. The variation in energy system contributions may partly be explained by differences in pre-exercise glycogen stores. Low pre-exercise glycogen stores have been found to reduce glycogen utilization for a given work output during endurance training [88], so the creatine phosphate and aerobic systems may contribute more during low-carbohydrate diets. Churchley et al. [89] found that a resistance training session induced 123 mmol/kg dry weight glycogen depletion under baseline circumstances, compared to 91 mmol/kg dry weight after prior glycogen depletion with bicycling; however, the difference was not statistically significant. Even assuming the lowest reported contributions of the aerobic and creatine phosphate systems, 20% and 16%, respectively, thereby assuming a 64% glycolytic contribution, and assuming a hypothetical but realistic strength training session energy expenditure of 500 kcal, this would require 80 g carbohydrate to fuel. Assuming 500 g glycogen storage in a typical athlete [3], this would theoretically amount to only 16% glycogen depletion.

Empirically, higher glycogen depletion levels have been reported. To the authors’ knowledge, the highest level of whole-muscle glycogen depletion after resistance training in the literature is 41%, or 39% if we exclude isokinetic exercise [4,5,8,18,52,79,89,90,91,92,93,94]. For example, Essén-Gustavsson and Tesch [90] found 28% quadriceps glycogen depletion in bodybuilders after five sets each of front squats, back squats, leg presses and leg extensions to failure at ~12 RM. Glycogen depletion only starts affecting neuromuscular functioning when levels are reduced to approximately 250–300 mmol/kg dry weight [10], which generally requires a depletion of over 40% from baseline, depending on the pre-exercise levels. Thus, resistance training workouts generally likely do not deplete enough glycogen to impair performance.

Overnight fasting should also not induce critically low glycogen levels: while liver glycogen content decreases after overnight fasting [95], intramuscular glycogen stores are not a substrate to maintain blood glucose concentrations and are therefore not depleted [96]. Given that the participants in the fasted acute studies were following their regular diet and did not exercise the evening prior to morning strength tests, muscle glycogen levels were likely not a limiting factor for strength training performance, at least in the lower-volume studies.

However, high-volume workouts may induce critically low glycogen levels in a subset of muscle fibers even when total muscle depletion levels are not critical. Hokken et al. [8] recently found that a total quadriceps glycogen depletion level of 38% after 12 sets of resistance training, excluding warm-up sets, was associated with approximately 50% subcellular depletion specifically within type II muscle fibers. The lowest quartile of intramyofibrillar, intermyofibrillar and subsarcolemmal glycogen stores decreased 72%, 60% and 62%, respectively. The depletion levels in the 25% most-depleted fibers were in the range where contractile functioning may be impaired. A notable limitation of Hokken et al. [8]’s study is that they estimated glycogen depletion from net utilization and did not factor in intra-exercise glycogen resynthesis. Thus, they likely overestimated the glycogen depletion. In comparison, Koopman et al. [91] found fiber-specific glycogen depletion was limited to 40% in the IIa and 44% in the IIx fibers after 16 sets quadriceps resistance training at 75% of 1 RM after an overnight fast in untrained individuals, although a direct comparison between the two studies is limited by their use of different methods to quantify glycogen depletion. Additionally, the participants in Hokken et al.’s [8] study were weightlifters and powerlifters with a relatively low pre-exercise glycogen storage level of 92 mmol/kg wet weight, despite being in a fed state. In comparison, Robergs et al. [5] reported 120 mmol/kg and Haff et al. [52] reported 150 mmol/kg baseline glycogen stores in strength trainees. It is possible Hokken et al.’s [8] weightlifters were unaccustomed to high-volume ‘bodybuilding style’ workouts, so 12 sets of resistance training may not induce critical glycogen depletion in trainees with more common levels of fed-state glycogen stores.

Nevertheless, critical depletion in type II muscle fibers after high-volume strength training could explain the trend in the literature that acute ergogenic effects of carbohydrate intakes are more prevalent, although not consistent, in workouts with more than 10 sets per muscle groups. In the four studies with test workouts with more than 10 sets per muscle group, there were two studies in favor of higher carbohydrate intakes [28,40], in addition to a trend in a third study [26], one study reporting no effect [33] and no studies favoring lower carbohydrate intakes. The study reporting no effect only measured repetitions in the last set [33], in contrast to all sets in the studies reporting (a trend for) benefits. Although we may expect the last set to be most affected by potential glycogen substrate depletion during a workout, measuring repetition volume during all sets may increase statistical power to detect potential benefits of higher carbohydrate intakes. In the 14 studies with test workouts with 10 or fewer sets per muscle group, there were only three studies in favor of higher carbohydrate intakes [29,39,42], nine studies reporting no effect [30,31,32,34,35,37,38,41,43] and two studies favoring lower carbohydrate intakes [27,36]. However, none of the studies favoring higher carbohydrate intakes had isocaloric control groups. Based on Mitchell et al. [48], in isocaloric conditions, carbohydrate intake may still not affect resistance training performance in recreational strength trainees up to 15 sets per muscle group even after a recent depletion workout.

The lack of isocaloric controls makes it impossible to determine whether the superior performance in some studies can be attributed to carbohydrate intake per se. Based on Naharudin et al. [41] there may also be a non-metabolic component to the ergogenic effects of a pre-workout meal, regardless of carbohydrate intake. These researchers found that a high-carbohydrate breakfast (1.5 g/kg) improved resistance training performance compared to drinking only water after an overnight fast; however, a flavor- and texture-matched placebo breakfast with only 29 kcal improved performance similarly. The sham breakfast also reduced hunger similarly. Thus, the feeling of having consumed something can be more important than carbohydrate intake per se. These results corroborate findings from a similar perception of breakfast experiment prior to a 30 min endurance performance trial [97]. However, another trial where the performance event was above 30 min and after a glycogen depletion protocol found that while the sham breakfast significantly improved performance over consuming just water, it did not raise performance to the level of a high carbohydrate intake [98], presumably because carbohydrate was actually a performance limiting substrate. While other strength training studies have attempted to match their placebos to their carbohydrate supplements in the form of liquids (e.g., [28,52]), none did so as rigorously as Naharudin et al. [41] with a semi-solid meal, double-blinding and by telling the participants their meal ‘contained energy’. A follow-up study from Naharudin et al. [99] compared two isocaloric breakfast meals, a semi-solid one vs. a liquid one. The semi-solid meal reduced hunger more and improved back squat repetition performance more than the liquid meal, suggesting hunger suppression can have a positive effect on resistance training performance. Since all studies finding benefits of higher carbohydrate intakes also had a higher energy intake, none of these effects may have necessarily been mediated by carbohydrate intake per se but rather by hunger suppression resulting in higher training efforts.

Moreover, other studies have found that carbohydrate mouth rinsing—without any actual carbohydrate consumption—can improve resistance training repetition performance compared to placebo [100] and that any placebo mouth rinse, regardless of carbohydrate content, can improve performance compared to water consumption [101]. A full review of the carbohydrate mouth rinsing literature is beyond the scope of this paper, but these findings cast doubt on the conventionally proposed metabolic role of pre-exercise feeding or rather taste experience. A predominantly psychological ergogenic effect of pre-workout feeding would also explain the complete lack of observed dose–response effect for carbohydrate intake in the literature, as well as by Krings et al. [29], since the sensation of having consumed something may be more important than carbohydrate consumption.

In conclusion, carbohydrate intake per se, independent of energy intake, is mechanistically and statistically unlikely to acutely affect resistance training performance in a fed state for workouts up to 10 sets per muscle group. Higher-volume workouts may require higher carbohydrate intakes to optimize performance, but there is a clear need for more isocaloric research with realistic placebos. However, given the uncertainty in the literature, based on Lynch [36] and Krings et al. [29], strength trainees may be advised to consume at least 15 g net carbohydrate and 0.3 g/kg protein within 3 hours pre-workout to optimize performance. If the workout involves more than 10 sets per muscle group, higher carbohydrate intakes might be warranted.

### 4.2. The Effect of Exercise-Induced Glycogen Depletion and Carbohydrate Manipulation on Acute Strength Performance

Three out of six studies that included a glycogen depletion session prior to strength tests found a positive effect of carbohydrate intake on performance [15,48,49,50,51,52]. Since the cycling depletion workouts were designed based on previous work to deplete glycogen stores by approximately 80% [102,103], it is likely that glycogen levels were below the threshold of impairing neuromuscular functioning after the depletion workouts. So, it is plausible the higher carbohydrate intakes helped bring glycogen stores back up to less limiting levels before the strength training test workouts.

Higher carbohydrate intakes were mainly beneficial in studies with short recovery times in between the depletion and the test workout. In the three studies with no more than four hours of recovery in between the workouts [49,50,52], two favored the higher-carbohydrate condition [49,50]. Only Haff et al. [52] found no benefit of carbohydrate intake compared to fasting for the workout output of three sets of isokinetic leg extensions, likely because they reported only 19% glycogen depletion after the prior depletion workout consisting of three sets of isokinetic leg extension strength testing and nine sets of squats. In the three studies with 48 h in between the two workouts [15,48,51], only one favored the higher-carbohydrate condition [15]. Glycogen resynthesis post-workout follows a biphasic recovery that is greatly enhanced by a high carbohydrate intake in the first four hours [80,104]. Since the second workout consisted of 19 sets of squats in the higher-carbohydrate condition in Haff et al. [49] and the participants in the lower-carbohydrate condition in Oliver et al. [50] were fasted, 2 to 4 h was likely not enough time to resynthesize glycogen back to levels that optimized performance during the second workout in either study. In the studies with 48 h rest in between the depletion workout and the test workout, only Leveritt and Abernethy [15] favored the higher-carbohydrate condition and this study was confounded by only performing the depletion workout in the lower-carbohydrate condition. Since the participants were only recreationally trained men, it is possible that the depletion workout interfered with performance due to muscle damage or otherwise incomplete recovery rather than glycogen depletion per se.

Two days may be enough time for complete glycogen resynthesis even on a low-carbohydrate diet. While high carbohydrate intakes are needed to resynthesize glycogen stores after exhaustive exercise as fast as possible [80], glycogen stores are partly autoregulated, as glycogen content is inversely related to glycogen synthase I activity, thereby allowing faster glycogen resynthesis after greater depletion, at least if substrate is available [104]. Even fasted, the insulin release of elevated blood glucose levels during exercise, combined with the ‘recycling’ of exercise-related lactate production via the Cori cycle, may allow a glycogen resynthesis rate of approximately 1.9 mmol/kg/h the first 2 h after resistance training [92] and likely faster after high-intensity endurance training [105]. A considerable amount of glucose, up to the daily requirements during starvation, can also be obtained from dietary fat intake or adipose tissue, via gluconeogenesis from the glycerol backbone of triglycerides [106]. Third, glucose can be produced from glucogenic amino acids, although this may only be desirable from a muscular anabolic point of view if the amino acids are consumed in excess of requirements for muscle protein synthesis [107]. Phielix et al. [108] measured the glycogen synthesis rates of sedentary and endurance-trained individuals in an overnight fasted state (with saline infusion) at 2.0 mmol/kg/h and 3.7 mmol/kg/h, respectively, using ^13^C/^31^P magnetic resonance spectroscopy. Assuming a resting glycogen storage level of 120 mmol/kg for strength trainees [5] and a depletion level of 80%, full resynthesis in 48 h requires a glycogen synthesis rate of 2 mmol/kg/h, which may thus be possible even in fasted conditions.

Just like in our other categories, none of the studies that favored the higher-carbohydrate conditions were isocaloric. The only isocaloric study in this category by Mitchell et al. [48] found no significant between-group differences in total training volume during 15 sets of resistance training at 15 RM to failure after a glycogen depletion workout followed by a carbohydrate intake of 32 g/day or 643 g/day for 48 h in strength-trained men. Lack of isocaloric comparisons is particularly confounding in studies with prior glycogen depletion workouts, because the extra carbohydrate/energy intake may not just aid glycogen resynthesis but also neuromuscular recovery. Glycogen depletion workouts are by nature exhaustive and generally a novel stimulus, so they have the potential to induce significant muscle damage and neuromuscular fatigue that may take over 48 h to recover from [109]. Additionally, glycogen depletion after a novel workout may not reflect the depletion level experienced by trainees habituated to the training stimulus and low-carbohydrate diets [13,110], although other research finds no habituation effects [111,112].

In conclusion, in addition to the previous recommendations, higher carbohydrate intakes are likely warranted for maximum performance when performing more than one glycogen-depleting workout per day. If there are only a few hours in between workouts, carbohydrate intakes up to 1.2 g/kg/h may be warranted to maximize glycogen resynthesis and subsequent performance.

### 4.3. The Effect of Short-Term Carbohydrate Manipulation on Acute Strength Training Performance

None of the seven short-term experiments found any positive effects on acute strength training performance following 2–7 days of a higher-carbohydrate intake compared to a lower-carbohydrate intake, neither in an isocaloric comparison [54] nor non-isocaloric comparisons [53,55,56,58,59,63]. For example, Moura et al. [59] observed that two days of a high-carbohydrate refeed (672 g/day) did not increase repetitions across 10 sets of leg presses in enhanced bodybuilders compared to 2 days of energy deficit with a lower carbohydrate intake (231 g/day).

One study by Sawyer et al. [58] favored the low-carbohydrate condition, but it was not randomized or counterbalanced. In this study, 31 strength trainees switched over from a 41% carbohydrate intake to a 5% carbohydrate intake (31 g/day) for a week before completing a battery of strength tests. After the low-carbohydrate diet, the participants had lost bodyweight due to a 15% lower total energy intake, yet handgrip strength, squat 1 RM and vertical jump height improved significantly compared to earlier testing during the higher-carbohydrate diet, although the squat strength improvement was only 0.9 kg. Bench press 1 RM, power and repetitions to failure and Wingate power output did not significantly change. Since the study design was not randomized, the improvements in the lower-carbohydrate condition may have been due to a familiarization or training effect rather than the diet, although the authors discounted this possibility based on the short study duration in comparison to the minimum 2 years of previous training experience of the participants. A familiarity effect would still imply that any ergogenic effects of a 234 g higher carbohydrate intake was considerably smaller than the effect of a single week of training in well-trained individuals (average 1 RM squat strength > 117 kg in a sample with 48% women). Another possible confounder was that protein intake increased significantly in the low-carbohydrate diet, although it still averaged 145 g/day during the higher-carbohydrate diet and the higher-carbohydrate diet had a higher energy intake (2537 vs. 2157 kcal).

The lack of any benefits of carbohydrate intake in the short-term studies can be understood based on glycogen metabolism. The participants mostly performed habitual training sessions with no more than 10 sets per muscle group, so it is likely that glycogen availability was not a limiting factor for their workouts. The participants had at least 24 h in between workouts in all the studies, which was likely sufficient for full glycogen replenishment. Even for the glycogen depletion amount of the study with the highest reported depletion in the literature after non-isokinetic resistance training, 39% or 46.6 mmol/kg [52], this would require an average glycogen resynthesis rate over 24 h of 1.9 mmol/kg/h for full recovery. This is on the lower end of the range of glycogen resynthesis rates (1.9–3.7 mmol/kg/h) found in fasted individuals [92,108]. In conclusion, strength trainees are unlikely to be limited by their carbohydrate intake to fuel habitual strength training workouts with no more than 10 sets per muscle group.

### 4.4. The Effect of Longer-Term Carbohydrate Diets and Strength Training on Changes in Strength Performance

A total of 16 out of 17 long-term studies found no significant benefits of carbohydrate manipulation on strength-training performance and strength development, including all isocaloric and isonitrogenous trials [61,62,63,64,65,75,76,77], all isocaloric non-isonitrogenous trials [66,67,71], the only isonitrogenous but non-isocaloric trial [74] and four out of five studies that equated neither protein nor energy intake between conditions [68,69,70,72,73]. One study favored the lower-carbohydrate condition [67] and one study favored the higher-carbohydrate condition [73].

The only study favoring the higher-carbohydrate condition by Vargas-Molina et al. [73] found greater increases in squat and bench press strength and fat-free mass (FFM) but not countermovement jump height in the higher-carbohydrate condition. However, despite the instruction to consume a similar energy intake, the lower-carbohydrate, ketogenic diet resulted in significantly lower self-reported energy intake (1710 kcal vs. 1980 kcal) and significantly more fat loss (1.1 kg loss vs. a non-significant 0.3 kg gain in the non-ketogenic group). Since none of the other nine ketogenic diets (<100 g/day) found attenuated strength development [61,62,63,65,67,68,69,70,72], the greater improvement in the higher-carbohydrate condition may have been due to the energy surplus in that group, vs. the energy deficit in the ketogenic group, rather than carbohydrate intake, although a recent meta-analysis did not find significantly detrimental effects of a daily 567 kcal energy deficit on strength development [113].

The only study favoring the lower-carbohydrate condition was by Rhyu and Cho [67], which found reduced anaerobic fatigue during a Wingate test in the lower-carbohydrate condition, but since none of the other performance tests favored the low-carbohydrate condition, including Wingate mean and peak power, the relevance of this lone finding is questionable.

Overall, the results indicate that carbohydrate intake does not have much, if any, effect on long-term resistance training performance in isonitrogenous and isocaloric conditions.

The carbohydrate-independent pathways by which the body can synthesize glucose may be sufficient in the context of sustainable training volumes and recovery times in between workouts. If these pathways were inadequate to cover all glucose requirements initially, they could become adequate over time, although this has yet to be observed in strength trainees. Volek et al. [114] compared the glycogen depletion and recovery of ultra-endurance athletes on a habitual 59% carbohydrate diet to those on a habitual 10% carbohydrate diet after a three-hour run. Ninety minutes pre-workout and immediately post-workout, the participants consumed an isocaloric (5 kcal/kg) and isonitrogenous (14% protein) shake with either 5% carbohydrate for the low-carbohydrate dieters or 50% carbohydrate for the high-carbohydrate dieters. Despite the large difference in habitual, pre-exercise and post-exercise carbohydrate intakes, resting glycogen concentrations were similar between groups, they decreased similarly during the workout (62–66%) and they recovered similarly over two hours back to 34–38% depletion. Other research finds habituated exercise in carbohydrate-restricted conditions may decrease reliance on glycogen, although all research so far is on endurance rather than strength training [115,116,117]. Furthermore, Phinney et al. [118], found that during a ketogenic, protein-sparing modified fast with endurance training, muscle glycogen levels stabilized at 69% of baseline after 6 weeks on the diet, compared to 57% of baseline after the first week. Moreover, there was a decrease the amount of glycogen depletion of their treadmill endurance test training from 16% at baseline to 13% in week 1 and unmeasurable levels in week 6, while the time to exhaustion increased by 55%.

Fifteen of the studies also estimated or measured fat-free-mass (FFM) or muscle thickness. Given the relationship between muscle size and strength, a between-group difference in muscle size could influence strength development [119], although the relation is also affected by multiple other factors [120]. Most of these studies found no significant between-group differences in these measures or body composition and thereby presumably muscle mass, in line with the lack of differences in strength development. However, five studies found a between-group difference favoring the higher-carbohydrate condition with either a decrease (0.7–1.7 kg) in the lower-carbohydrate group [61,72,73] compared to an increase in the higher-carbohydrate groups (0.5–0.8 kg) or a lower increase in FFM (1.4 kg vs. 3.4 kg and 0.6 kg vs. 2.2 kg) in the lower-carbohydrate condition [74,76], compared to only one study favoring the lower-carbohydrate condition [65]. However, the aforementioned Vargas-Molina et al. [73] study finding greater increases in strength in the higher carbohydrate condition was confounded by significantly lower energy intake in the lower-carbohydrate condition, as was Rozenek et al.’s study [74] with 4339 kcal vs. 2597 kcal in the higher- vs. lower-carbohydrate conditions, respectively. LaFountain et al.’s study [72] was probably similarly confounded, as it had ad libitum diets without instructions regarding energy intake, resulting in significant fat loss in the ketogenic diet group (−5.9 kg) but not the control diet (−0.6 kg). Greene et al.’s study [61] was isocaloric based on self-reported food logs and the authors noted that the loss of FFM in this study may have largely been water and glycogen, combined with a previously reported overestimation of FFM loss in carbohydrate-restricted athletes. Lack of significant loss of contractile muscle tissue would explain why neither basal metabolic rate nor performance measures significantly differed between groups. Similarly, in Paoli et al.’s [76] study, the participants did not report significantly different energy intakes between groups; however, again only the ketogenic diet group lost a significant amount of fat (−1.4 kg) and strength performance and resting metabolic rate did not significantly differ between groups, so the greater FFM gain may in the Western diet group may have been attributable largely to water and glycogen losses in the ketogenic group. Glycogen levels have been shown to influence FFM measurements [121]. Notably, Wilson et al. [65] demonstrated that one week of carbohydrate reintroduction after a 10-week ketogenic diet increased FFM by 4.8%, while no change was observed in the traditional high-carbohydrate group. Thus, the overall literature does not support lower-carbohydrate diets impair muscular development when accounting for energy intake and glycogen storage levels. However, from a practical standpoint it seems more challenging to consume a target energy surplus in ketogenic diets, possibly due to the appetite suppressive effect of ketogenic diets [122]. Muscular development is significantly affected by energy balance [113]. Thus, strength development may eventually be attenuated on a low-carbohydrate diet if less muscle size is gained in periods longer than the three months of the included studies, as argued by others [123,124]. Nevertheless, based on our systematic review of the available literature, carbohydrate intake does not seem to have much, if any, effect on strength development up to three months duration.

### 4.5. Conclusions and Practical Applications

The majority of research, including every isocaloric comparison, did not find higher carbohydrate intakes improve strength training performance, either acutely or over the course of a strength training program, compared to lower carbohydrate intakes. There is also evidence that the positive effects of higher carbohydrate intakes in comparison to nothing or less filling controls can be non-metabolic, possibly mediated by hunger suppression and subsequently greater exercise efforts. However, subgrouping the studies shows that carbohydrate manipulation may be beneficial in certain contexts, namely otherwise fasted training, workouts with more than 10 sets per muscle group and bi-daily workouts. These contexts merit further study with sensory-matched placebos and isocaloric, isonitrogenous control groups. Mechanistically, resistance training workouts up to 10 sets per muscle group are unlikely to sufficiently deplete glycogen stores below the threshold of impairing neuromuscular functioning. Glycogen resynthesis after such depletion may be complete within 24 h even on low carbohydrate intakes via carbohydrate-independent pathways, especially in trainees habituated to the training in low-carbohydrate conditions. However, the existing literature on direct glycogen measurements is limited, so future research should study how localized glycogen compartments are affected by training volume and how they affect exercise performance. Since our findings are based on research on adults, future research should also investigate if these findings can be extrapolated to individuals below 18 years of age or over 60 years of age.

Overall, our findings indicate conventional high-carbohydrate intake recommendations of 4–10 g/kg/day may be excessive for the performance of strength trainees, such as bodybuilders, powerlifters and Olympic weightlifters. Based on the inconclusive evidence and potential for benefits but not harm, strength trainees are advised to consume at least 15 g carbohydrates and 0.3 g/kg protein within 3 hours of their training sessions. If the workout contains eleven or more sets per muscle group or there is another high-intensity workout planned that day for the same musculature, higher carbohydrate intakes up to 1.2 g/kg/h may be warranted to maximize glycogen resynthesis in between workouts. Future research is needed to validate these dosages.

## Figures and Tables

**Figure 1 nutrients-14-00856-f001:**
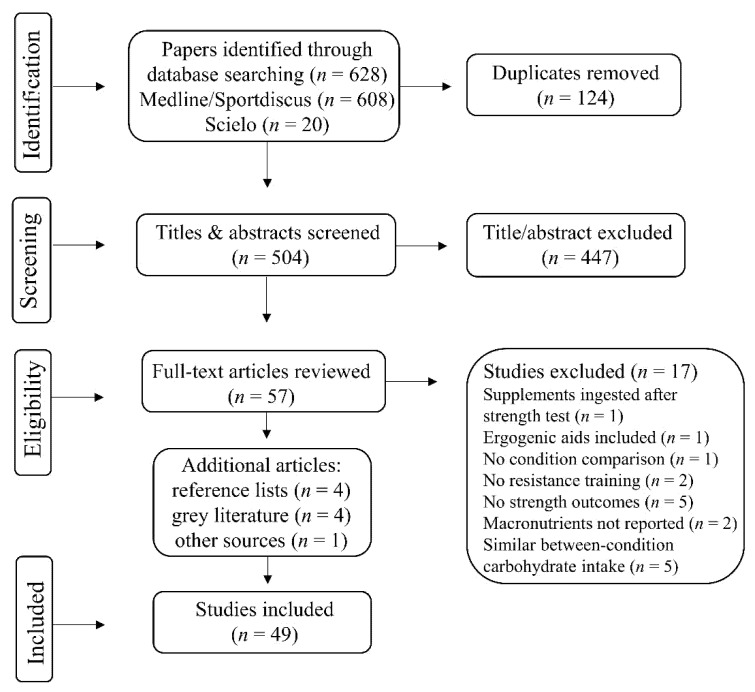
Flow chart of the study selection process.

**Figure 2 nutrients-14-00856-f002:**
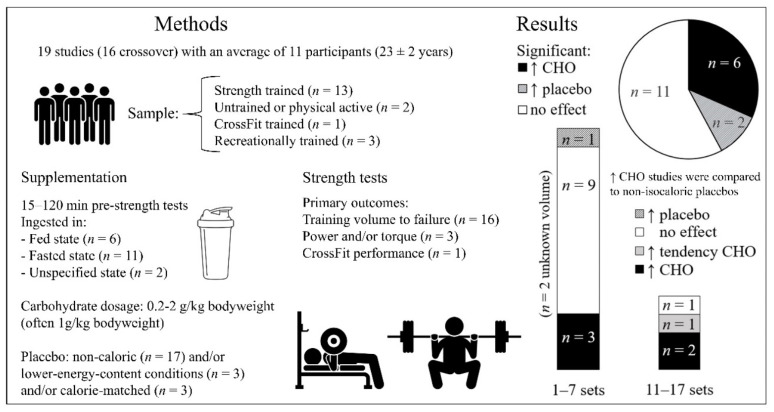
Acute carbohydrate intake and effect on strength performance. ↑: “Greater performance for”.

**Figure 3 nutrients-14-00856-f003:**
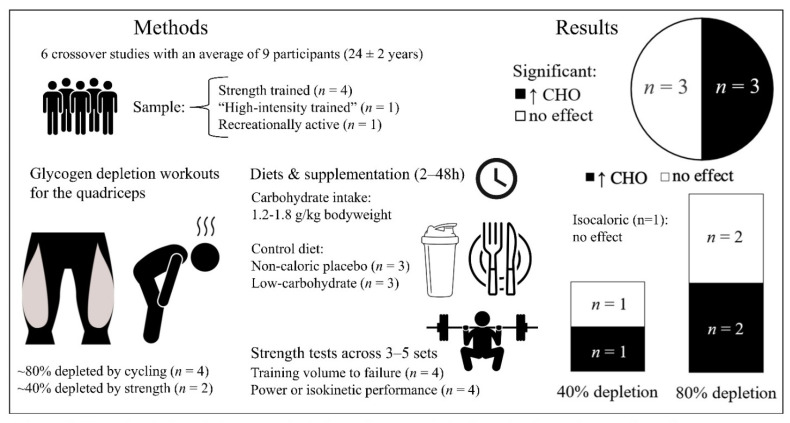
Exercise-induced glycogen depletion prior to carbohydrate intake and strength performance tests. ↑: “Greater performance for”.

**Figure 4 nutrients-14-00856-f004:**
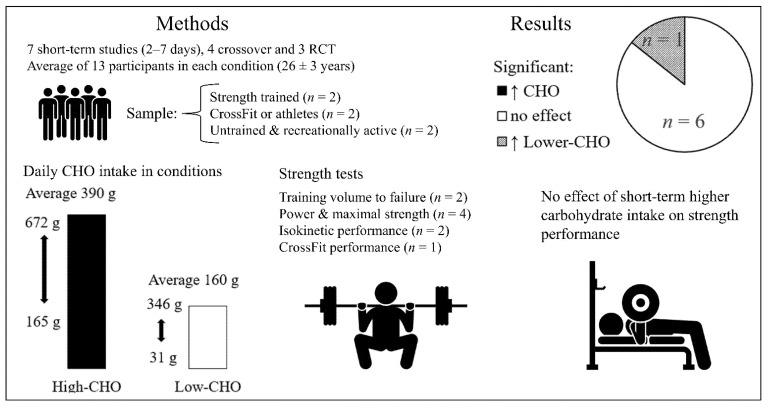
Short-term carbohydrate manipulation on strength performance. ↑: “Greater performance for”.

**Figure 5 nutrients-14-00856-f005:**
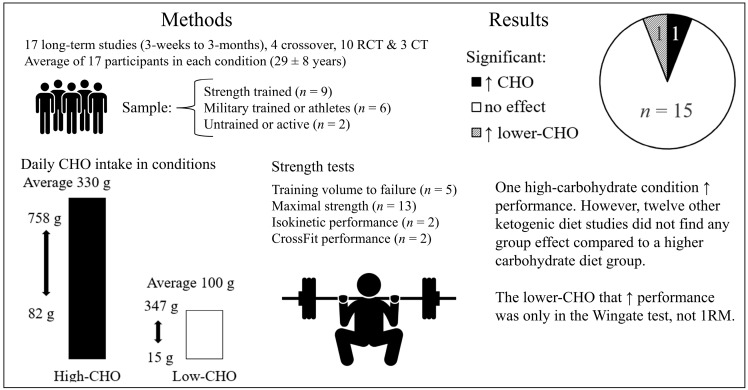
Long-term carbohydrate manipulation on strength adaptations. ↑: “Greater performance for”.

**Table 1 nutrients-14-00856-t001:** Summary of study quality assessment in acute-, glycogen depletion- and short-term studies.

	Criterion	*n*	%
Study quality	1. Eligibility criteria specified	21	66
	2. Randomization specified	5	16
	3. Allocation concealment	30	94
	4. Groups similar at baseline	32	100
	5. Blinding of assessor (for at least one key outcome)	16	50
Study reporting	6a. Outcome measures assesses in 85% of participants	30	94
	6b. Adverse events reported	1	3
	8a. Between-group statistics reported—primary	32	100
	8b. Between-group statistics reported—secondary	32	100
	9. Points measures and measures of variability reported	31	97
	12. Exercise volume and energy expenditure	29	91

1 = criteria met; 0 = criteria not met; *n* = number of studies meeting criteria: % = percentage of studies meeting criteria. Thirty-one studies in total.

**Table 2 nutrients-14-00856-t002:** Summary of study quality assessment in long-term studies.

	Criterion	*n*	%
Study quality	1. Eligibility criteria specified	16	94
	2. Randomization specified	0	0
	3. Allocation concealment	12	71
	4. Groups similar at baseline	15	88
	5. Blinding of assessor (for at least one key outcome)	1	6
Study reporting	6a. Outcome measures assesses in 85% of participants	14	82
	6b. Adverse events reported	7	41
	6c. Exercise attendance reported	13	76
	7. Intention-to-treat analysis	7	41
	8a. Between-group statistics reported—primary	17	100
	8b. Between-group statistics reported—secondary	17	100
	9. Points measures and measures of variability reported	17	100
	11. Relative exercise intensity remained constant	16	94
	12. Exercise volume and energy expenditure	5	29

1 = criteria met; 0 = criteria not met; *n* = number of studies meeting criteria: % = percentage of studies meeting criteria. Fourteen studies in total.

**Table 3 nutrients-14-00856-t003:** The acute effect of carbohydrate ingestion on strength training performance.

Study	Design and Population	Training Protocol and Performance Outcomes	Carbohydrate (CHO) Intakes	Fasted or Fed	Results
Baty et al. [32]	RCT: Healthy untrained men (*n* = 32), carbohydrate-protein group vs. placebo group.	**Training**: 7 exercises (high pull, lat pull-down, standing overhead press, knee extension, leg curl, leg press and bench press) with the two first sets as 8 RM, and the third set with the same load as set 2 but until voluntary failure. **Outcome**: total weight lifted (kg) during the last set per exercise and weight lifted scaled per lean body mass multiplied by the number of repetitions completed during the last set per exercise.	**CHO**: 0.59 g/kg (44 g [6.2%] and 1.5% protein). **Placebo**: non-caloric. **Timing**: 355 mL 30 min prior to exercise, 177 mL immediately before and after the fourth exercise.	Fasted (12 h overnight).	No significant differences between conditions in weight lifted the last set or total training volume (total load CHO-PRO: 534 ± 80 kg vs. placebo: 556 ± 82 kg; weight scaled per lean body mass × repetitions CHO-PRO: 93 ± 17 vs. placebo: 92 ± 21).
Dalton et al. [33]	RCT: Strength-trained subjects (*n* = 22), carbohydrates (*n* = 8) vs. placebo (*n* = 8) vs. control (*n* = 6) in caloric deficit	**Training**: lower-body exercises (squat, leg press and knee extension) and bench press at 60–80% of 10 RM, 5-sets per exercise. **Outcome**: last set of knee extension and bench press 80% of 1 RM to failure.	**CHO**: 1 g/kg beverage supplement. **Placebo**: non-caloric supplement. **Control**: no supplement. **Timing**: 30 min before testing.	Overnight fasted.	No significant differences in repetitions to failure between conditions (knee extension CHO: 17 ± 1, placebo: 17 ± 2, control: 17 ± 2; bench press CHO: 17 ± 2, placebo: 17 ± 2, control: 16 ± 3).
Fairchild et al. [27]	Counterbalanced crossover: Strength-trained men (*n* = 11) and women (*n* = 6), carbohydrate vs. placebo.	**Training**: one set of 3 RM knee extensions in an isokinetic dynamometer, and again after 5, 15, 30, 45, 60, 75 and 90 min. **Outcome**: peak and average isokinetic torque.	**CHO**: 1.1 g/kg (75 g). **Placebo**: non-caloric supplement. **Timing**: after the first baseline 3 RM.	Fasted (>12 h overnight).	There was no interaction effect but when adjusting for baseline values a significant main effect between conditions were observed where the CHO condition resulted in a decline (~2%-points) and maintenance in average and peak torque, respectively, compared to an increase (~4–5%-points) in both for placebo.
Fayh et al. [34]	Crossover: Strength-trained subjects (*n* = 8), carbohydrate vs. placebo.	**Training**: seven exercises (bench press, lat pulldown, rear deltoid, barbell curl, hammer curl, leg press and squat) with three sets with an intensity of 70% 1 RM to failure. **Outcome**: total training volume (repetitions × sets × load).	**CHO**: 1 g/kg (84 g) of maltodextrin beverage supplement. **Placebo**: non-caloric supplement. **Timing**: 15 min before training.	Fed (2 h pre).	No significant differences in total training volume between conditions (CHO: 12,944 ± 2548 kg vs. placebo: 12,876 ± 2025 kg).
Haff et al. [28]	Crossover: Strength-trained men (*n* = 8), carbohydrate vs. placebo.	**Training**: 16 sets of 10 repetitions with isokinetic knee extension and flexion **Outcome**: total and average work (J) across all sets, peak and average isokinetic torque (Nm) across all sets.	**CHO**: 1.0 g/kg prior to exercise and 0.51 g/kg during exercise (143 g in total). **Placebo**: non-caloric. **Timing**: before exercise and after set 1, 6 and 11.	Fed (3 h pre).	Significant greater total work (CHO: 24 ± 2 J, placebo: 22 ± 2 J), average work (CHO: 1.5 ± 0.1 J, placebo: 1.4 ± 0.5 J), and average torque per set (CHO: 105 ± 8 Nm, placebo: 98 ± 8 Nm) in knee extension in the CHO condition. No differences were observed between conditions in peak torque in the knee extension or any of the measurements for the knee flexors.
Krings et al. [29]	Crossover: Strength-trained men (*n* = 7), carbohydrates, amino acids and electrolytes vs. amino acids and electrolytes (placebo).	**Training**: explosive high-intensity training and resistance training: hang clean at 50–70% 1 RM, front squat at 45–90% 1 RM, box jumps, dumbbell bench press and barbell bent-over row at 60–73% 1 RM, barbell reverse lunge at 55–70% 1 RM, single-arm shoulder press at 65–70% 1 RM, dumbbell biceps curl and dumbbell overhead triceps extension at 60% 1 RM. Three to seven sets for all exercises. **Outcome**: last set to failure in dumbbell bench press, barbell bent-over row, dumbbell biceps curl and dumbbell overhead triceps extension. Sprints, jump distance, overhead medicine ball throws and agility tests.	**CHOs**: 15, 30 or 60 g/h corresponding to a 3, 6 and 12% solution. In addition to 5.5 g amino acids (AA) and electrolytes. **Placebo**: 5.5 g AA and electrolytes. **Timing**: before exercise and every 15 min during exercise, total 5 dosages.	Fasted (at least 10 h overnight).	No significant differences in total repetitions between CHOs and placebo, but 15 g/h > 60 g/h. For the bench press, all CHO groups outperformed placebo without dose-response. No significant differences for the other three exercises, two jumps or four run times, except 60 g/h > placebo for the 27-m sprint.
Kulik et al. [35]	Counterbalanced crossover: Strength-trained men (*n* = 8), carbohydrate vs. placebo.	**Training**: sets of five repetitions at 85% 1 RM until subjects could no longer squat to parallel, failed to do a repetition every 8 s, or reached voluntary failure, with 3-min rest between sets. **Outcome**: repetitions and sets to failure, in addition to volume load (load × sets × repetitions) and total work (kJ).	**CHO**: 0.3 g/kg (28 g). **Placebo**: non-caloric. **Timing**: before and after every other set of 5 repetitions.	Fed (3 h pre).	No significant differences between conditions in repetitions and sets to failure or volume load and total load (repetitions CHO: 20 ± 15 vs. placebo: 20 ± 13, sets CHO: 4 ± 3 vs. placebo: 4 ± 3, volume load CHO: 2929 ± 2220 kg vs. placebo: 2773 ± 1951 kg, work CHO: 30 ± 22 kJ vs. placebo: 29 ± 20 kJ).
Lambert et al. [26]	Crossover: Strength-trained men (*n* = 7), carbohydrate vs. placebo.	**Training**: knee extensions at 80% of 10 RM, first set was performed with 10 repetitions, then subsequent sets were performed until one failed to perform 7 repetitions in a single set. **Outcome**: repetitions and sets to failure.	**CHO**: 1 g/kg before exercise, and 0.17 g/kg dosages during exercise (97 or 125 g in total). **Placebo**: non-caloric. **Timing**: before exercise, an after set 5, 10 and 15.	Relatively fed (4-h pre).	No significant difference in repetitions and sets to failure between the conditions. However, there was a tendency for more repetitions (149 ± 16 vs. 129 ± 12, *p* = 0.067) and sets (17 ± 2 vs. 14 ± 2, *p* = 0.056) in the CHO condition.
Laurenson and Dubé [30]	Crossover: Strength-trained men (*n* = 10), carbohydrates vs. placebo.	**Training**: Seven sets of squat and bench press (60% 1 RM), first 6 with a predetermined number of repetitions. **Outcome**: last set was performed to repetition failure where the total volume (kg load × repetitions) and peak power output was measured.	**CHO**: 0.43 g/kg (36 g and 12 g of protein). **Placebo**: non-caloric. **Timing**: two dosages, 12 and 26 min into exercise.	Fasted (8–10 h).	Significantly more total bench press volume in the CHO condition (921 ± 365 vs. 783 ± 332). However, no differences was observed in total squat volume (CHO: 1009 ± 433 vs. 909 ± 472, *p* = 0.1) or peak power for either bench press or squat.
Lynch [36]	Crossover: Strength-trained men (*n* = 15), carbohydrates vs. high-protein (including carbohydrates, protein and fat).	**Training**: high-intensity resistance training for 2 min (overhead push-press, dumbbell push-press, squats and dumbbell push-ups) for as many rounds as possible. **Outcome**: performance tests 2 h after the workout; agility T-test, push-up (repetitions to failure), and 40-yard sprint.	**CHO**: A total of 0.84 g/kg (68 g). **High-protein**: 40 g protein, 11 g of carbohydrate and 6 g fat (isocaloric to CHO). **Timing**: within 5 min of completing the first workout.	Not specified.	No significant difference between conditions in agility T-test, push-ups to failure or sprint. However, analyzing all three performance variables simultaneously yielded a significant greater effect of the high-protein condition compared to the carbohydrate condition.
Maroufi et al. [44]	Crossover: Male CrossFit athletes (*n* = 8), carbohydrate-protein supplement in two ratios (2:2 or 3:1) vs. placebo.	**Training**: two 15–17 min CrossFit workouts. **Outcome**: repetitions to failure.	**CHO-protein (ratio 3:1)**: 67.5 g CHO and 22.5 g protein. **CHO-protein (ratio 2:2)**: 45 g CHO and 45 g protein. **Placebo**: non-caloric. **Timing**: 1 h and immediately before testing.	Fasted (overnight)	No significant difference between conditions in repetitions to failure (3:1 ratio 341 ± 56, 2:2 ratio 366 ± 61, placebo 346 ± 65).
Naharudin et al. [39]	Counterbalanced crossover: Strength-trained men (*n* = 16), breakfast vs. a water-only breakfast.	**Training**: Four sets to failure with squat and bench press at 90% of 10 RM. **Outcome**: repetitions to failure.	**CHO**: A total of 1.5 g/kg (116 g), standardized breakfast meal, ~20% of estimated energy needs. **Control**: water only. **Timing**: 2 h before testing.	Fasted (~10 h overnight).	Significantly more repetitions to failure in the CHO condition for squat (68 ± 14 vs. placebo: 58 ± 11, effect size [ES] = 0.98) and bench press (40 ± 5 vs. placebo: 38 ± 5, ES = 1.06).
Naharudin et al. [41]	Counterbalanced crossover: Strength-trained men (*n* = 22), breakfast vs. placebo-breakfast vs. water-only.	**Training**: Four sets to failure with squat and bench press at 90% of 10 RM. **Outcome**: repetitions to failure.	**CHO**: A total of 1.5 g/kg (117 g), standardized breakfast meal, 496 kcal. **Placebo**: semi-solid, 29 kcal with low-energy flavored squash and water. **Control**: water only. **Timing**: ~2 h before testing.	Fasted (10–13 h overnight).	Significantly more repetitions to failure in the CHO and placebo breakfast conditions in the squat exercise (CHO: 44 ± 10, placebo: 43 ± 10, water-only: 38 ± 10), but not during bench press (CHO: 39 ± 7, placebo: 38 ± 7, water-only: 37 ± 7). While there was no significant difference in repetitions completed in the CHO- vs. the placebo condition.
Raposo et al. [37]	Counterbalanced crossover: Strength-trained women (*n* = 13), carbohydrates vs. placebo.	**Training**: Five sets with 75% of 1 RM of bench press and 85% of 1 RM for leg press. **Outcome**: repetitions to failure and total volume (sets × repetitions × load) for each exercise and all exercises together.	**CHO**: A total of 1 g/kg (81 g). **Placebo**: non-caloric. **Timing**: A total of 1 h before exercise.	Fasted (overnight).	No significant differences between conditions in repetitions to failure and training volume (repetitions bench press, CHO: 45 ± 11 vs. 45 ± 10; leg press, CHO: 112 ± 59 vs. 98 ± 38. Training volume bench press, CHO: 1451 ± 414 vs. 1430 ± 387; leg press, CHO: 19,960 ± 13,477 vs. 17,103 ± 8927).
Rountree et al. [38]	Crossover: Strength-trained men (*n* = 8), carbohydrates vs. placebo.	**Training**: Five rounds of wall throws with a 9 kg medicine ball, box jumps, sumo deadlift high pulls with 34 kg, push presses with 34 kg for as many repetitions as possible within 1 min, and rowing ergometer at maximum effort for 1 min. **Outcome**: repetitions to failure and caloric expenditure during rowing.	**CHO**: A total of 0.2 g/kg (16 g). **Placebo**: non-caloric. **Timing**: before exercise and during the training session (6 total dosages of 2.7 g each).	Fasted (10–12 h overnight).	No significant differences between conditions in repetitions to failure (total repetitions CHO: 279 vs. placebo: 272) and caloric expenditure during 1 min all out rowing (kilocalories CHO: 42 vs. placebo: 45).
Santos et al. [42]	Crossover: Strength-trained men (*n* = 8), carbohydrates vs. placebo.	**Training**: one set of bench-press, 70% of 1 RM to failure. **Outcome**: repetitions to failure.	**CHO**: A total of 0.27 g/kg (20 g). **Placebo**: non-caloric. **Timing**: 1 h before training.	Not specified.	Significantly more repetitions in the CHO condition (13 ± 2 vs. 11 ± 2).
Smith et al. [31]	Crossover: Strength-trained men (*n* = 13), carbohydrates vs. carbohydrates + BCAA vs. BCAA vs. placebo.	**Training**: barbell bench press, landmine bent-over row, barbell incline press, and landmine close-grip row. All exercises were performed with 5 sets to failure at 65% of 1 RM. **Outcome**: repetitions to failure.	**CHO**: A total of 0.44 g/kg (36 g). **CHO + BCAA**: A total of 36 g and 7.5 g BCAA. **BCAA**: A total of 7.5 g **Placebo**: non-caloric. **Timing**: the total dosage was distributed to be ingested before and after warm-up, and after the last set of each exercise.	Fasted (10 h overnight).	No significant time × treatment interactions for any exercise for repetition performance. However, there was a treatment effect for CHO + BCAA compared to the other treatments, but it was confounded by an order effect. Additionally, close-grip row repetitions to failure were greater in the CHO-BCAA condition compared to the other conditions.
Welikonich [40]	RCT: Recreational strength-trained men (*n* = 27), carbohydrates vs. CHO-protein vs. placebo.	**Training**: multiple sets with leg press of 8–10 repetitions at 70% of 1 RM until fatigue (unable to reach 8 repetitions) **Outcome**: total number of repetitions in addition to sets to failure, measured as total training volume (load × repetitions × sets).	**CHO**: A total of 0.81 g/kg (~60 g), 0 g PRO **CHO-PRO**: A total of 0.65 g/kg (~50 g) CHO, ~14 g PRO **Placebo**: non-caloric (15 kcal) **Timing**: A total of 15 min before training (~30 g) and between every other set (in total ~30 g).	Fed (standardized liquid meal 5 h pre).	Significantly more repetitions in the CHO and CHO-PRO condition (CHO: 136 ± 55/36, respectively) vs. placebo (90 ± 15). However, no difference was observed between groups in total volume of work (CHO: 28,052 ± 19,198 kg vs. CHO-PRO: 24,836 ± 9737 vs. placebo: 15,934 ± 3276 kg (*p* = 0.13).
Wilburn et al. [43]	Crossover: Recreational strength-trained men (*n* = 10), carbohydrates vs. placebo.	**Training**: Four sets of leg press at 70% of 1 RM to failure. **Outcome**: repetitions to failure.	**CHO**: A total of 2 g/kg (180 g). **Placebo**: non-caloric. **Timing**: 30 min before training.	Fed (3 h pre, instructed not to change dietary habits).	No significant differences between conditions (total repetitions CHO: 52 ± 7, placebo: 54 ± 8, *p* = 0.80).

**Table 4 nutrients-14-00856-t004:** The effect of exercise-induced glycogen depletion and carbohydrate manipulation on acute strength performance.

Study	Design and Population	Training Protocol and Performance Outcomes	Carbohydrate- (CHO) Intakes	Fasted or Fed	Results
Haff et al. [49]	Counterbalanced crossover: Strength-trained men (*n* = 6), carbohydrate vs. placebo.	**Glycogen depleting workout**: Five sets of 10 repetitions in squats (65% 1 RM), speed squats (45% 1 RM) and 1-legged squat (10% 1 RM). **Training**: A total of 4 h after the first workout, session two was performed; as many sets of 10 squats with 55% of 1 RM as possible (to failure) with a 3-min rest interval. **Outcome**: completion of as many sets with 10 repetitions as possible.	Both conditions received a standardized high-carbohydrate (~1.2 g/kg) lunch 2.5 h prior to strength tests (~825 kcal). **CHO**: 1.2 g/kg/h during the morning session, 0.38 g/kg/h during the 4 h recovery period between workouts, while a non-specified dosage was provided every second set (total carbohydrate intake not specified). **Placebo**: non-caloric. **Timing**: morning, recovery period and during exercise.	Fed (2.5 h pre).	Significantly more repetitions and sets to failure in the CHO condition (total repetitions CHO: 199 ± 115 vs. placebo: 131 ± 67, total sets CHO: 19 ± 12 vs. placebo: 11 ± 7). There was no significant difference in the total work performed between conditions, but a tendency for a difference in favor of the CHO condition (336 ± 217 vs. placebo: 224 ± 114, *p* = 0.066).
Haff et al. [52]	Crossover: Strength-trained men (*n* = 8), carbohydrate vs. placebo.	**Training**: Three sets of 10 repetitions of knee extension and flexion in an isokinetic dynamometer, pre and post depletion workout: 3 sets of 10 repetitions of squats (65% 1 RM), speed squats (45% 1 RM) and 1-legged squat (10% 1 RM). **Outcome**: total and average work (J) across sets, peak and average isokinetic torque (Nm) before and after the training bout.	**CHO**: A total of 1 g/kg pre and 0.3 g/kg 3 × during the depletion workout (163 g in total) **Placebo**: non-caloric. **Timing**: before exercise and 3 drinks during.	Fed (3 h pre).	No significant differences in the isokinetic measurements between conditions. (Glycogen levels were reduced by ~41% in the placebo condition and ~27% in the carbohydrate condition.)
Leveritt and Abernethy [15]	Crossover: Recreationally active men (*n* = 5) and women (*n* = 1); first tested strength, then performed a glycogen depletion workout 5 days later and 2 days of a low carbohydrate diet (~100 g per day) prior to strength tests again.	**Glycogen depletion workout**: cycling at 75% of VO^2^ max for 1 h, 3 min rest, followed by four 1 min bouts at 100% of VO^2^ max with 3-min rest intervals. **Outcome**: Three sets of isoinertial squat at 80% of 1 RM performed until failure with 3-min rest-intervals, in addition to 5 repetitions with isokinetic knee extension torque at five different contraction speeds.	**Lower carbohydrate**: ~1884 kcal 1.21 g/kg (90 g carbohydrates). **Control diet**: Not reported.	Not specified.	Significantly more repetitions at set 1 and 2 during squats in the control diet group compared to the low-carbohydrate diet (set 1: CHO: 18 ± 8, control: 12 ± 5, set 2: CHO: 14 ± 6, control: 10 ± 4. No significant difference was observed in set 3 (CHO: 10 ± 7, control: 11 ± 4) or in torque during knee extensions.
Mitchell et al. [48]	Counterbalanced crossover: Strength-trained men (*n* = 11); high-carbohydrate and a low-carbohydrate condition for 48 h after glycogen depletion.	**Glycogen depletion workout**: cycling at 70% of VO^2^ max for 1 h, followed by 1-min sprints at 115% of VO^2^ max with 1-min rest-intervals. **Outcome**: after the 48 h diet period; five sets at 15 RM of squats, leg presses and knee extensions to failure. Performance was quantified as total volume lifted.	**Lower carbohydrate**: 3094 kcal 0.4 g/kg CHO/protein/fat 32/226/230 g **Higher carbohydrate**: 3206 kcal 7.7 g/kg CHO/protein/fat 643/84/33 g	Not specified.	No significant differences in total training volume between groups (high-carbohydrate: ~15,800 kg, low-carbohydrate: ~15,500 kg).
Oliver et al. [50]	Crossover: Strength-trained men (*n* = 16), two carbohydrate conditions with different molecular weight and osmolarity vs. placebo.	**Glycogen depletion workout**: cycling for 60 min at 70% VO^2^ max, followed by six 1-min sprints at 120% of maximal aerobic power. **Training**: A total of 2 h after cessation of the first workout, session two was performed; squats at 75% of 1 RM, five sets of 10 as explosive as possible. **Outcome**: average power output, force and velocity across all squat sets.	**CHO**: 1.2 g/kg (106 g), high molecular weight and low osmolarity (HMW), and low molecular weight and high osmolarity (LMW). **Placebo**: non-caloric. **Timing**: after the glycogen depletion bout (2 h before strength training).	Fasted (overnight 12 h).	The carbohydrate conditions achieved significantly greater average power outputs and movement velocities than placebo, but the differences between groups in total training volume or average force output were insignificant or of ‘trivial’ magnitude.
Symons and Jacobs [51]	Counterbalanced crossover: Men (*n* = 8) with experience with high-intensity training; glycogen depleted the knee extensors with cycling, then subjects followed two diets the next two days; low carbohydrate diet or a mixed diet.	**Outcome**: knee extension electrically evoked isometric muscle force, voluntary isometric strength and isokinetic total work across all repetitions (J), peak and average torque from 50 maximal unilateral knee extensions, in addition to muscle fatigue (average torque of the last three contractions divided by the peak torque).	**Lower carbohydrate**: A total of 3000 kcal 140 g, 1.8 g/kg/day (19%) carbohydrates. **Higher carbohydrate**: Not reported.		No significant differences between groups in any of the performance measurements.

**Table 5 nutrients-14-00856-t005:** The effect of short-term (2–7 days) carbohydrate manipulation on acute strength performance.

Study	Design and Population	Training Protocol and Performance Outcomes	Diet	Results
**Isocaloric studies**				
Dipla et al. [54]	Counterbalanced crossover: Recreationally active women (*n* = 10); control diet or a high-protein lower-carbohydrate diet for 1 week each.	**Outcome**: handgrip strength and four sets of 16 maximal repetitions (120° per seconds) with isokinetic knee extensors and flexors contractions. Isokinetic peak torque determined by three maximal efforts, and muscle fatigue as the percentage reduction in work produced in the last set relative to the first set.	**Lower carbohydrate**: A total of 1305 kcal CHO/protein/fat 99/131/43 g **Higher carbohydrate**: A total of 1315 kcal CHO/protein/fat 179/53/43 g	No significant differences in peak torque or muscle fatigue between groups.
**Non-isocaloric, protein non-equated studies**				
Escobar et al. [59]	RCT: Male (*n* = 7) and female (*n* = 11) CrossFit athletes (*n* = 18); high-carbohydrate (*n* = 9) or a control group (*n* = 9). Subjects consumed their regular diet for 5 days, then the carbohydrate group increased carbohydrate intake to 6–8 g/kg/day.	**Training**: CrossFit workouts on day 6 and 7. **Outcome**: number of repetitions performed in a 12-min CrossFit workout on day 1, 5 and 9.	**Lower carbohydrate**: 1846 kcal CHO/protein/fat 213/105/64 g **Higher carbohydrate**: 2938 kcal CHO/protein/fat 428/129/79 g	No significant differences between groups in number of repetitions during CrossFit training.
Hatfield et al. [55]	Counterbalanced crossover: Strength-trained men (*n* = 8); diet with 50% or 80% of calories from carbohydrates for 4 days.	**Outcome**: Four sets of 12 squat jumps at 30% 1 RM. Power output and total work (J) was measured.	**Lower carbohydrate**: A total of 50% carbohydrates **Higher carbohydrate**: A total of 80% carbohydrates	No significant differences between groups in any of the performance measurements.
Kreider et al. [56]	RCT: Male athletes (*n* = 14); carbohydrate supplement group (4 g/kg/day) or a placebo group for 7 days.	**Training**: One to two intensive hockey training sessions per day. **Outcome**: vertical jump, 1 RM bench press and leg press.	**Lower carbohydrate**: A total of 2398 kcal 346 g (58%) carbohydrates **Higher carbohydrate**: A total of 3685 kcal 628 g (68%) carbohydrates	No significant differences between groups in any of the performance measurements.
Meirelles et al. [57]	RCT: Sedentary women (*n* = 24); 500–800 kcal deficit conventional diet (*n* = 12) or ad libitum very low carbohydrate diet (VLCD, *n* = 12) for 1 week.	**Outcome**: Three sets of 15 maximal effort knee extensions in the concentric phase at a velocity of 60°/s. Peak torque, average power, set total work (J), and total work across all sets were measured.	**Lower carbohydrate**: A total of <40 g carbohydrates per day **Higher carbohydrate:** CHO/protein/fat 48/22/30% 165 g carbohydrates	No significant differences between groups in any of the isokinetic measurements.
Moura et al. [59]	RCT: Enhanced male bodybuilders (*n* = 11); moderate energy deficit (*n* = 6) or severe energy deficit (*n* = 5) with acute strength tests in the fourth diet week after two days of low-calorie lower-carbohydrate intake and then after 2 days of refeed with higher-carbohydrate intake.	**Training**: followed their usual resistance training with five sessions per week. **Outcome**: total repetitions to failure, 10 sets of leg press at 70% 1 RM with 10 RM and 30 s rest-intervals.	**Lower carbohydrate**: Moderate energy deficit: 2968 kcal CHO/protein/fat 227/295/98 g Severe energy deficit: 2507 kcal CHO/protein/fat 235/271/54 **Higher carbohydrate**: Refeed after moderate energy deficit: 4039 kcal CHO/protein/fat 687/151/76 g Refeed after severe energy deficit: 3715 kcal CHO/protein/fat 655/116/70 g **Combined moderate and energy deficit groups**: **Lower carbohydrate**: A total of 2758 kcal CHO/protein/fat 231/284/78 g **Higher carbohydrate refeed**: A total of 3892 kcal CHO/protein/fat 672/135/73 g	No significant differences between groups in number of repetitions.
Sawyer et al. [58]	Crossover: Strength-trained men (*n* = 16) and women (*n* = 15); habitual diet for 7 days, and then a carbohydrate restricted diet for 7 days.	**Training**: required to complete a 1-week resistance trained log, so likely continued their usual training. **Outcome**: handgrip strength, bench press and back squat 1 RM, bench press peak power, followed by repetitions to failure, in addition to countermovement vertical jump height and peak power output from a 30 s Wingate.	**Lower carbohydrate**: A total of 2157 kcal CHO/protein/fat 31/201/137 g **Higher carbohydrate**: A total of 2537 kcal CHO/protein/fat 265/145/100 g	Significantly greater handgrip strength, squat 1 RM, and vertical jump height in the carbohydrate restricted condition compared to the control condition, with no difference in the other measurements.

**Table 6 nutrients-14-00856-t006:** The effect of longer-term carbohydrate diets and strength training on changes in strength performance.

Study	Design and Population	Strength Training and Performance Outcomes	Diet	Results
**Isocaloric, isonitrogenous studies**				
Greene et al. [61]	Crossover: Intermediate to elite male (*n* = 9) and female (*n* = 5) powerlifters and Olympic weightlifters; low-carbohydrate ketogenic diet or to continue their usual ad libitum diet, in a random order for 3 months (*n* = 12 completed).	**Training**: subjects were instructed to maintain their normal training. **Outcome**: 1 RM for one or all of the subjects’ competition lifts.	**Lower carbohydrate**: A total of 2072 kcal CHO/protein/fat 41/119/159 g **Higher carbohydrate**: A total of 2058 kcal CHO/protein/fat 223/119/79 g	No significant difference between groups in changes in 1 RM.
Gregory et al. [62]	RCT: CrossFit athletes (*n* = 27) of both genders; low-carbohydrate ketogenic diet (*n* = 12) group or to maintain their normal dietary intake (control, *n* = 15) for 6 weeks.	**Training**: Four CrossFit training sessions per week. **Outcome**: changes in countermovement vertical jump height and standing long jump length and time-performance during a standardized CrossFit workout.	**Lower carbohydrate**: A total of 1581 kcal CHO/protein/fat 44/92/115 g **Higher carbohydrate**: A total of 1747 kcal CHO/protein/fat 187/80/73 g	No significant differences between groups in changes of any of the performance measurements.
Meirelles and Gomes [63]	CT: Overweight (≥25 BMI) but strength-trained males and females (*n* = 21) self-selected to follow a low-carbohydrate (*n* = 12) or a conventional/habitual diet (*n* = 9) for 8 weeks.	**Training**: full-body resistance training was performed three times per week, two sets of 11 exercises with 8–10 RM and 2-min rest-intervals. **Outcome**: 10 RM in the biceps pulldown, triceps pushdown and leg press.	**Lower carbohydrate**: A total of 1566 kcal CHO/protein/fat 83 g carbohydrates 1.5 g/kg/day protein **Higher carbohydrate**: A total of 1459 kcal CHO/protein/fat 171 g carbohydrates 1.6 g/kg/day of protein	No significant differences between groups in changes in any of the 10 RM tests.
Michalski et al. [75]	Female and male CrossFit athletes (*n* = 22); first 2 weeks of their usual diet, then a low-carbohydrate ketogenic diet for 4 weeks.	**Training**: maintaining their usual training. **Outcome**: as many repetitions as possible within a 17 min CrossFit workout.	**Lower carbohydrate**: A total of 2807 kcal CHO/protein/fat 33/125/238 g **Higher carbohydrate:** A total of 2565 kcal CHO/protein/fat 290/118/104 g	No significant differences between groups in CrossFit repetition performance.
Paoli et al. [76]	RCT: Male bodybuilders (*n* = 19); low-carbohydrate ketogenic diet (*n* = 9) or western diet (*n* = 10) for 8 weeks.	**Training**: maintaining their usual strength training (3–4 sessions per week). **Outcome**: 1 RM squat and bench press.	**Lower carbohydrate**: A total of 3444 kcal CHO/protein/fat 44/216/264 g **Higher carbohydrate:** A total of 3530 kcal CHO/protein/fat 488/223/79 g	No significant differences between groups in 1 RM changes.
Van Zant et al. [64]	Crossover: Strength-trained males (*n* = 6); high-carbohydrate or a moderate-carbohydrate diet for 3 weeks.	**Training**: maintaining their usual strength training. **Outcome**: knee- extension and flexion peak torque and total work performed during two sets of 30 isokinetic contractions, bench press 1 RM and bench press repetitions to failure at 80% 1 RM.	**Lower carbohydrate**: CHO: A total of 4.2 g/kg/day (~347 g) **Higher carbohydrate:** CHO: A total of 6.3 g/kg/day (~520 g)	No significant differences between groups in changes of any of the performance measurements.
Vidić et al. [77]	RCT: Strength-trained males (*n* = 18); low-carbohydrate ketogenic diet group (*n* = 9) or a non-ketogenic diet group (*n* = 9) for 8 weeks.	**Training**: resistance training was performed four times per week as a split-routine, unspecified load, three sets and 6–12 repetitions per set. **Outcome**: A total of 1 RM squat and bench press.	**Lower carbohydrate**: A total of 2156 kcal CHO/protein/fat 27/108/180 g **Higher carbohydrate**: A total of 2191 kcal CHO/protein/fat 82/110/158 g	No significant differences between groups in 1 RM changes.
Wilson et al. [65]	RCT: Strength-trained men (*n* = 25); low-carbohydrate ketogenic diet group (*n* = 13) or a western diet group (*n* = 12) for 10 weeks (carbohydrates were then reintroduced in the ketogenic group in the 11th week).	**Training**: resistance training was performed three times per week as a split-routine, 65–95% of 1 RM, three to four sets per exercise and 1–15 repetitions per set. **Outcome**: bench press and back squat 1 RM, and 10 s Wingate cycle sprint (peak power).	**Lower carbohydrate**: 2617 kcal CHO/protein/fat 31/135/217 g **Higher carbohydrate**: A total of 2545 kcal CHO/protein/fat 317/131/84 g	No significant differences between groups in changes of any of the performance measurements.
**Isocaloric, non-isonitrogenous studies**				
De Oliveira et al. [71]	RCT: Male military police students (*n* = 16); protein supplement (4 g/kg/day, *n* = 8) or a carbohydrate supplement (225 g, *n* = 8) group for 8 weeks.	**Training**: resistance training three × per week, 80% 1 RM for eight repetitions × five sets. Exercises were arm curls, preacher curls, overhead triceps and lying down triceps extension. **Outcome**: maximal strength (1 RM) for all exercises, and peak torque from five repetitions of isokinetic elbow flexion and extension.	**Lower carbohydrate**: A total of 3710 kcal CHO/protein/fat 338/297/112 g **Higher carbohydrate**: A total of 3767 kcal CHO/protein/fat 581/130/100 g	No significant differences between groups in changes of 1 RM or peak torque.
Kreider et al. [66]	RCT: Obese women (*n* = 221); high-carbohydrate (*n* = 92) or a high-protein, low-carbohydrate diet (*n* = 129) for 10 weeks (diets consisted of 1200 kcal the first week, then 1600 kcals the next 9 weeks).	**Training**: supervised whole-body circuit resistance training three × per week. **Outcome**: bench press 1 RM and repetitions to failure at 70% of 1 RM.	**Lower carbohydrate**: 1411 kcal CHO/protein/fat 123/102/57 g **Higher carbohydrate**: 1379 kcal CHO/protein/fat 183/62/46 g	No significant differences between groups in changes in repetitions to failure or 1 RM.
Rhyu and Cho [67]	RCT: Male taekwondo athletes (*n* = 20); ketogenic diet group (*n* = 10) or a non-ketogenic diet group (*n* = 10) for 3 weeks in a 25% caloric deficit.	**Training**: resistance training and taekwondo training were performed 6 days per week. **Outcome**: grip strength, back strength and repetitions of sit ups performed in 60 s, 100 m sprint, Wingate peak and mean power and fatigue index and standing broad jump distance.	**Lower carbohydrate**: A total of CHO/protein/fat 4/41/55% 22 g CHO per day **Higher carbohydrate**: A total of CHO/protein/fat 40/30/30%	No significant differences between groups in changes in any performance outcomes, except for a significantly lower (better) anaerobic fatigue index during the Wingate test in the ketogenic group.
**Non-isocaloric, protein equated studies**				
Rozenek et al. [74]	RCT: Active males (*n* = 46); high-carbohydrate diet (*n* = 25) or a control group (*n* = 21) for 8 weeks.	**Training**: resistance training was performed four times per week as a 2-split routine, eight repetitions for four sets. **Outcome**: 1 RM in bench press, leg press, lat pull-down and in total.	**Lower carbohydrate**: A total of 2597 kcal CHO/Protein/Fat 337/107/84 g **Higher carbohydrate**: A total of 4339 kcal CHO/Protein/Fat 758/109/87 g	No significant differences between groups in changes of maximal strength.
**Non-isocaloric, non-isonitrogenous studies**				
Agee [68]	RCT: Male powerlifters (*n* = 12); ad libitum low-carbohydrate ketogenic diet (*n* = 4) or to maintain their habitual diet, control group (CON) (*n* = 8) for 6 weeks.	**Training**: resistance training was performed four times per week as a 2-split routine, 4–12 repetitions forthree to five sets. **Outcome**: 1 RM in bench press, squat and deadlift.	**Lower carbohydrate**: A total of 1918 kcal CHO/protein/fat: 107/136/106 g **Higher carbohydrate:** A total of 2862 kcal CHO/protein/fat: 268/166/121 g	No significant differences between groups in changes of maximal strength.
Kephart et al. [69]	CT: Male (*n* = 9) and female (*n* = 3) CrossFit athletes (*n* = 12); self-selected to either continue their normal diet (*n* = 5) or follow a ketogenic diet (*n* = 7) for 12 weeks.	**Training**: continued CrossFit workouts (ketogenic diet group completed 27 workouts, whereas the control completed 20 workouts). **Outcome**: back squat and power clean 1 RM, one set of push-up repetitions to failure and 400-m running time.	**Lower carbohydrate**: 1948 kcal CHO/protein/fat: 15/89/170 g **Higher carbohydrate:** Not reported.	No significant differences between groups in changes of any of the performance measurements.
LaFountain et al. [72]	CT: Healthy military men (*n* = 25) and women (*n* = 4); self-selected to follow an ad libitum ketogenic diet or to continue their normal mixed diet for 12 weeks.	**Training**: supervised full-body resistance training 2 × per week. three to four sets, 4–12 repetitions at 60–95% 1 RM. **Outcome**: countermovement vertical jump power, 1 RM squat and bench press, 10 sprint intervals and obstacle course performance.	**Lower carbohydrate**: <50 g/day carbohydrates **Control diet:** >40% carbohydrates	No significant differences between groups in changes of any of the performance measurements.
Paoli et al. [70]	Crossover: Elite male gymnasts (*n* = 8); ad libitum very-low-carbohydrate ketogenic diet for the first 30 days, and then 30 days with a Western diet 3 months later.	**Training**: instructed to continue their normal training schedule of approx. 30 h per week. **Outcome**: One set of pushups, pull ups, dips, hanging straight and bodyweight leg raises until failure, in addition to squat- and countermovement jumps.	**Lower carbohydrate**: A total of 1973 kcal CHO/Protein/Fat: 22/201/120 g **Higher carbohydrate**: A total of 2276 kcal CHO/protein/fat: 264/84/97 g	No significant differences between groups in changes of any of the performance tests.
Vargas-Molina et al. [73]	RCT: Strength-trained women (*n* = 21); non-ketogenic diet (*n* = 11) or a ketogenic diet (*n* = 10) for 8 weeks.	**Training**: supervised 2-split resistance training four times per week (strength, hypertrophy and muscle endurance phases: 3–25 repetitions × 3 sets). **Outcome**: 1 RM squat and bench press, and countermovement jump height.	**Lower carbohydrate**: A total of 1710 kcal CHO/protein/fat: 39/115/122 g **Higher carbohydrate:** A total of 1980 kcal CHO/protein/fat: 282/97/51 g	Significantly greater increase in changes of 1 RM for squat and bench press in the non-ketogenic diet group (10- and 3.3 kg difference, respectively), with no group differences in CMJ performance.

## Data Availability

No new data were created or analyzed in this study. Data sharing is not applicable to this article.

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
