# Peer review of "The Effect of Carbohydrate Intake on Strength and Resistance Training Performance: A Systematic Review"

_nutrients, 2022, doi:10.3390/nu14040856_

Round 1

Reviewer 1 Report

The authors have conducted a well designed systemic review of literature to understand the effects of carbs on performance in adult athletes.

The manuscript is well written and I congratulate the authors for their findings.

There are minor comments that the authors may want to consider. Please see the attached file.

Author Response

Thank you for your review of our manuscript. We have replied to your comments in detail in the attachment. In short, we implemented all your suggestions and fixed the table numbering. Please let us know if you have any additional comments.

Reviewer 2 Report

This is a systematic review that evaluated The effect of carbohydrate intake on strength and resistance training performance. The topic is relevant and current and the article is well written. However, some minor review is needed.

  • Line 90- Please see "Implementing the 27 PRISMA 2020 Statement items for systematic reviews in the sport and exercise medicine, musculoskeletal rehabilitation and sports science fields: the PERSiST (implementing Prisma in Exercise, Rehabilitation, Sport medicine and SporTs science) guidance" . In this article, there are registration options that replace prospero.
  • Line 96- Please make an appendix with the search strategy used in each database
  • Line 106- Please update the search. According Cochrane "Searches for all the relevant databases should be rerun prior to publication, if the initial search date is more than 12 months (preferably six months) from the intended publication date "
  • Methodology: Are letters, congress abstracts included? Are all languages included? Please, Specify exclusion criteria
  • Fig 1. Please specify reasons for exclusion
  • Line 146-155 Please explain further the reasons for excluding these items. It is not clear.
  • Line 156- it is recommended that the risk of bias assessment be performed by two reviewers independently. 
  • line 166-167 - The authors mention that they have added 6 additional papers from reference lists and 5 from authors’ previous knowledge on the topic. Does this not demonstrate that the search strategy used was not good since known articles were not retrieved by the search?
  • Excellent figures with a summary of the results. 
  • Please, item 4.1 is too long. Please be more succinct
  • Please review the conclusion of items 4.1 and 4.2 and the overall conclusion of the manuscript "Based on the available evidence, strength trainees are advised to consume at least 15 g carbohydrates and 0.3 g/kg protein within three hours of their training sessions. If the workout contains eleven or more sets per muscle group or there's another high-intensity workout planned that day for the same musculature, higher carbohydrate intakes up to 1.2 g/kg/h may be warranted to maximize glycogen resynthesis in between workouts."
    Considering that, due to the diversity of protocols of the selected studies, it was not possible to perform meta-analysis, how did the authors determine these recommendations? Do you think it is correct to reach this conclusion with such a small number of studies and with different protocols? I suggest that instead of providing these recommendations, authors should discuss this diversity and demonstrate what changes in protocols are necessary to arrive at more reliable results. Another possibility is to perform a meta-analysis of the 8 acute studies that have more homogeneous protocols (strength trained, fasted state, non-caloric placebo) and from the Meta-analysis, observe which amount of CHO provided greater performance.

Author Response

Thank you for your review of our manuscript. We will address your comments point by point below.

  • Line 90- Please see "Implementing the 27 PRISMA 2020 Statement items for systematic reviews in the sport and exercise medicine, musculoskeletal rehabilitation and sports science fields: the PERSiST (implementing Prisma in Exercise, Rehabilitation, Sport medicine and SporTs science) guidance" . In this article, there are registration options that replace prospero.

Thank you for making us aware of this. We adjusted the text in the paper to highlight that this is a limitation and that there are other options that
future reviews should consider: “The present systematic review followed Preferred Reporting Items for Systematic Reviews and Meta-Analyses (PRISMA) guidelines [22]. We did not pre-register the present review since the protocol did not fulfill the requirements for preregistration at Prospero, which state that they do not accept reviews assessing sports performance as an outcome. However, in retrospect, there are other options that we could have used (Pieper & Rombey 2022).”

  • Line 96- Please make an appendix with the search strategy used in each database

Please find attached the requested appendix.

  • Line 106- Please update the search. According Cochrane "Searches for all the relevant databases should be rerun prior to publication, if the initial search date is more than 12 months (preferably six months) from the intended publication date "

We debated doing this previously ourselves, as it took a long time to create the manuscript, so we have now redone the search up to the year 2022. We found 5 additional papers from 2021: 1 acute, 1 short-term and 3 long-term, in addition to a short-term abstract. All these papers aligned with our main findings, so they did not alter our conclusions. Making these updates was the reason we needed an extension until today to reply to you.

  • Methodology: Are letters, congress abstracts included? Are all languages included? Please, Specify exclusion criteria

We have specified this in the updated manuscript: "Papers in all languages were eligible. Congress abstracts were eligible for inclusion but presented in their own sections, not among the main findings. Letters were not included."

  • Fig 1. Please specify reasons for exclusion

Good point. We have updated the figure to specify the exclusion reasons.

  • Line 146-155 Please explain further the reasons for excluding these items. It is not clear.

We have elaborated as follows: "..were excluded for the acute-, glycogen depletion- and short-term study categories, as they did not apply
in an acute context where participants do not follow a long-term diet and training intervention."

  • Line 156- it is recommended that the risk of bias assessment be performed by two reviewers independently. 

We agree. TB now went through the assessment as well. This led to minor changes in the scoring of some studies.

  • line 166-167 - The authors mention that they have added 6 additional papers from reference lists and 5 from authors’ previous knowledge on the topic. Does this not demonstrate that the search strategy used was not good since known articles were not retrieved by the search?

We debated this as well. The studies we did not find in the search were mostly grey literature or somewhat oddly classified with keywords such as 'intense training', 'physical training' or 'training program' rather than resistance/strength training/exercise. In the new search, we added a grey literature search in Google Scholar with the same keywords. This turned up 4 of the 5 studies. We also managed to find 2 studies in the references of other studies.

  • Excellent figures with a summary of the results.

Thank you.

  • Please, item 4.1 is too long. Please be more succinct

The section is very long, but are there any specific sections you feel could be excluded? We feel this section is paramount for the understanding of the research question. There are multiple topics to touch on: the psychological aspects, lack of isocaloric comparisons, the glycogen depletion level, etc. Considering we're not exceeding a word limit, we're uncomfortable with removing relevant information.

  • Please review the conclusion of items 4.1 and 4.2 and the overall conclusion of the manuscript "Based on the available evidence, strength trainees are advised to consume at least 15 g carbohydrates and 0.3 g/kg protein within three hours of their training sessions. If the workout contains eleven or more sets per muscle group or there's another high-intensity workout planned that day for the same musculature, higher carbohydrate intakes up to 1.2 g/kg/h may be warranted to maximize glycogen resynthesis in between workouts."
    Considering that, due to the diversity of protocols of the selected studies, it was not possible to perform meta-analysis, how did the authors determine these recommendations? Do you think it is correct to reach this conclusion with such a small number of studies and with different protocols? I suggest that instead of providing these recommendations, authors should discuss this diversity and demonstrate what changes in protocols are necessary to arrive at more reliable results. Another possibility is to perform a meta-analysis of the 8 acute studies that have more homogeneous protocols (strength trained, fasted state, non-caloric placebo) and from the Meta-analysis, observe which amount of CHO provided greater performance.

We have added the line: "Based on the inconclusive evidence and potential for benefits but not harm, strength trainees are advised... " Afterwards, we also added the line: "Future research is needed to validate these dosages." We do feel it is useful to recommend something concretely for practitioners, as this is the Practical Applications section. The recommendations are based on the potential for benefits of added protein (e.g. Lynch) and the absence of any discernible dose-response but possible detrimental effects of training fasted. So we settled on the recommendation of 15 g as the lowest point above which there is no dose-response effect based on Krings et al. Specific future research directions for scientists are provided in multiple other sections, including earlier in the Conclusions.

Please let us know if you have any follow-up comments.
